# GMMSeg: Gaussian Mixture based Generative Semantic Segmentation Models

**Chen Liang**[1,3*†], **Wenguan Wang**[2*], **Jiaxu Miao**[1], **Yi Yang**[1]

[1]CCAI, Zhejiang University    [2]ReLER, AAII, University of Technology Sydney    [3]Baidu Research

https://github.com/leonnnop/GMMSeg

## Abstract

Prevalent semantic segmentation solutions are, in essence, a dense *discriminative* classifier of $p(\textit{class}|\textit{pixel feature})$. Though straightforward, this *de facto* paradigm neglects the underlying data distribution $p(\textit{pixel feature}|\textit{class})$, and struggles to identify out-of-distribution data. Going beyond this, we propose GMMSeg, a new family of segmentation models that rely on a dense *generative* classifier for the joint distribution $p(\textit{pixel feature}, \textit{class})$. For each class, GMMSeg builds Gaussian Mixture Models (GMMs) via Expectation-Maximization (EM), so as to capture class-conditional densities. Meanwhile, the deep dense representation is end-to-end trained in a discriminative manner, *i.e.*, maximizing $p(\textit{class}|\textit{pixel feature})$. This endows GMMSeg with the strengths of both generative and discriminative models. With a variety of segmentation architectures and backbones, GMMSeg outperforms the discriminative counterparts on three closed-set datasets. More impressively, without any modification, GMMSeg even performs well on open-world datasets. We believe this work brings fundamental insights into the related fields.

## 1 Introduction

Semantic segmentation aims to explain visual semantics at the pixel level. It is typically considered as a problem of pixel-wise classification, *i.e.*, assigning a class label $c \in \{1, \cdots, C\}$ to each pixel data $x$. Under this regime, deep-neural solutions are naturally built as a combination of two parts (Fig. 1(a)): an encoder-decoder, *dense feature extractor* that maps $x$ to a high-dimensional feature representation $\boldsymbol{x}$, and a *dense classifier* that conducts $C$-way classification given input pixel feature $\boldsymbol{x}$. Starting from the first end-to-end segmentation solution – fully convolutional networks (FCN)[1], researchers leave the classifier as *parametric softmax*, and fully devote to improving the dense feature extractor for learning better representation. As a result, a huge amount of FCN-based solutions[2–5] emerged and their state-of-the-art was further pushed forward by recent Transformer[6]-style algorithms[7–10].

From a probabilistic perspective, the softmax classifier, supervised by the cross-entropy loss together with the feature extractor, directly models the class probability given an input, *i.e.*, posterior $p(c|\boldsymbol{x})$. This is known as a ***discriminative*** classifier, as the conditional probability distribution discriminates directly between the different values of $c$[11]. As discriminative classifiers directly find the classification rule with the smallest error rate, they often give excellent performance in downstream tasks, and hence become the *de facto* paradigm in segmentation. Yet, due to the discriminative nature, softmax-based segmentation models suffer from several limitations: **First**, they only learn the decision boundary between classes, without modeling the underlying data distribution[11]. **Second**, as only one weight vector is learned per class, they assume unimodality for each class[12, 13], bearing no within-class variation. **Third**, they learn a prediction space where the model accuracy deteriorates rapidly away

---

[*]Equal contributions.

[†]Work partly done during an internship at Baidu Research.

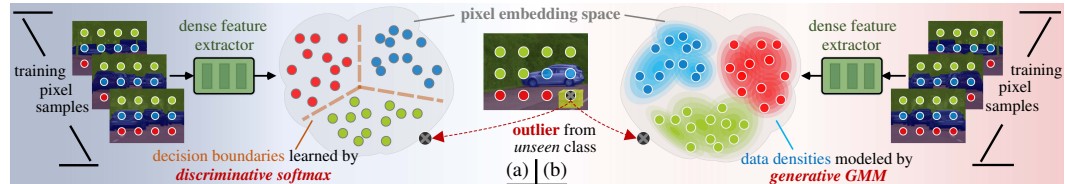

Figure 1: (a) Existing softmax based discriminative regime only learns decision boundaries on the pixel embedding space. (b) Our GMMSeg models pixel feature densities via generative GMMs.

from the decision boundaries [14] and thus yield poorly calibrated predictions [15], struggling to recognize out-of-distribution data [16]. The first two limitations may hinder the expressive power of segmentation models, and the last one challenges the adoption of segmentation models in decision-critical tasks (*e.g.*, autonomous driving) and motivates the development of anomaly segmentation methods [17–19] (which, however, rely on pre-trained discriminative segmentation models).

As an alternative of discriminative classifiers, ***generative*** classifiers first find the joint probability $p(\boldsymbol{x}, c)$, and use $p(\boldsymbol{x}, c)$ to evaluate the class-conditional densities $p(\boldsymbol{x}|c)$. Then classification is conducted using Bayes rule. Numerous theoretical and empirical comparisons [20, 21] between these two approaches have been initiated even before the deep learning revolution. They reach the agreement that generative classifiers have potential to overcome shortcomings of their discriminative counterparts, as they are able to model the input data itself. This stimulates the recent investigation of generative (and discriminative-generative hybrid [22, 23]) classifiers in trustworthy AI [24–27] and semi-supervised learning [22, 23], while the discriminative classifiers are still dominant in most downstream tasks.

In light of this background, we propose a GMM based segmentation framework – GMMSeg – that addresses the limitations of current discriminative solutions from a generative perspective (Fig. 1(b)). Our work not only represents a novel effort to advocate generative classifiers for end-to-end segmentation, but also evidences the merits of generative approaches in a challenging, dense classification task setting. In particular, we adopt a separate mixture of Gaussians for modeling the data distribution of each class in the feature space, *i.e.*, class-conditional feature densities $p(\boldsymbol{x}|c)$. During training, GMM classifier is *online* optimized by a momentum version of (Sinkhorn) EM [28] on large-scale, so as to ensure its generative nature and synchronization with the evolving feature space. Meanwhile, the feature extractor is *end-to-end* trained with the discriminative (cross-entropy) loss, *i.e.*, maximizing the conditional likelihood $p(c|\boldsymbol{x})$ derived with the generative GMM, so as to enable expressive representation learning. In this way, GMMSeg smartly learns generative classification with end-to-end discriminative representation in a compact and collaborative manner, exploiting the benefit of both generative and discriminative approaches. This also greatly distinguishes GMMSeg from most existing GMM based neural classifiers, which are either discriminatively trained [12, 29–31] or trivially estimate a GMM in the feature space of a pre-trained discriminative classifier [19, 32, 33].

GMMSeg has several appealing facets: **First**, with the hybrid training strategy – online EM based classifier optimization and end-to-end discriminative representation learning, GMMSeg can precisely approximate the data distribution over a robust feature space. **Second**, the mixture components make GMMSeg a structured model that well adapts to multimodal data densities. **Third**, the distribution-preserving property allows GMMSeg to naturally reject abnormal inputs, without neither architectural change (like [34–37]) nor re-training (like [38–40]) nor post-calibration (like [17, 18, 41–46]). **Fourth**, GMMSeg is a *principled* framework, fully compatible with modern segmentation network architectures.

For thorough examination, in §4.1, we approach GMMSeg on several representative segmentation architectures (*i.e.*, DeepLab$_{V3+}$[47], OCRNet [48], UperNet [49], SegFormer [7]), with diverse backbones (*i.e.*, ResNet [50], HRNet [51], Swin [52], MiT [7]). Experimental results demonstrate GMMSeg even outperforms the softmax-based discriminative counterparts, *e.g.*, **0.6% – 1.5%**, **0.5% – 0.8%**, and **0.7% – 1.7%** mIoU gains over ADE$_{20K}$ [53], Cityscapes [54], and COCO-Stuff [55], respectively. Furthermore, in §4.2, we validate our approach on anomaly segmentation. Without any modification, our Cityscapes-trained GMMSeg model is directly tested on Fishyscapes Lost&Found [56] and Road Anomaly [36] datasets, and outperforms all hand-tailored discriminative competitors.

To our best knowledge, GMMSeg is the first semantic segmentation method that reports promising results on both closed-set and open-world scenarios by using a single model instance. More notably, our impressive results manifest the advantages of generative classifiers in a large-scale real-world setting. We feel this work opens a new avenue for research in this field.

## 2 Related Work

**Semantic Segmentation.** Since the seminal work of FCN [1], deep-net segmentation solutions are typically built in a dense classification fashion, *i.e.*, learning dense representation and categorization end-to-end. By directly adopting discriminative softmax for categorization, FCN-style solutions put focus on learning expressive dense representation; they modify the FCN architecture from various aspects, such as enlarging the receptive field [2,3,47,57–60], modeling multi-scale context [48,59, 61–77], investigating non-local operations [4,5,78–84], and exploring hierarchical information [85–87]. With a similar goal of sharpening representation, later Transformer-style solutions [7–9,88, 89] empower attentive networks with, for instance, local contiguity [7] and multi-level feature aggregation [9,10]. Two very recent attentive models [90,91] formulate the task in an alternative form of *mask classification*, however, still relying on discriminative softmax.

From the discussion above, we can find that *current prevalent segmentation solutions are in essence a pixel-wise, discriminative classifier*, which only learns decision boundaries between classes in the pixel feature space [14,92], without modeling the underlying data distribution. In contrast, our GMMSeg tackles the task from a *generative* viewpoint. GMMSeg deeply embeds generative optimization of GMMs into end-to-end dense representation learning, so as to comprehensively describe the class-aware knowledge [93,94] in a discriminative feature space. GMMSeg is partly inspired by [13,95], that also probe data structures via intra-class clustering. However, the dense classification in the two works are achieved via non-parametric, nearest centroid retrieving – still a discriminative model. In [96], though data density is estimated (as a mixture of vMF distributions [97]), it is only used as a supervisory signal for dense embedding learning, and the final prediction is still made by a discriminative classifier – $k$-NN. Our work represents the first step towards formulating (closed-set) semantic segmentation within a generative neural classification framework.

**Discriminative *vs* Generative Classifiers.** Generative classifiers and discriminative classifiers represent two contrasting ways of solving classification tasks [24]. Basically, the generative classifiers (such as Linear Discriminant Analysis and naive Bayes) learn the class densities $p(\boldsymbol{x}|c)$, while the discriminative classifiers (such as softmax) learn the class boundaries $p(c|\boldsymbol{x})$ without regard to the underlying class densities. In practical classification tasks, softmax discriminative classifier is used exclusively [24], due to its simplicity and excellent discriminative performance. Nonetheless, generative classifiers are widely agreed to have several advantages over their discriminative counterparts [21, 98], *e.g.*, accurately modeling the input distribution, and explicitly identifying unlikely inputs in a natural way. Driven by this common belief, a surge of deep learning literature [14,99–101] investigated the potential (and the limitation) of generative classifiers in adversarial defense [25,26,102–104], explainable AI [24], out-of-distribution detection [27,105], and semi-supervised learning [22,23,99].

As GMMs can express (almost) arbitrary continuous distributions, it has been adopted in many neural classifiers [23,106]. However, most of these GMM classifiers are discriminative models [12,14,29,30] that are trained 'discriminatively' (*i.e.*, maximizing posteriors $p(c|\boldsymbol{x})$). In GMMSeg, the GMM is purely optimized via EM (*i.e.*, estimating class densities $p(\boldsymbol{x}|c)$) while the deep representation is trained via gradient backpropagation of the discriminative loss. Thus the whole GMMSeg is a hybrid of generative GMM and discriminative representation, getting the best of two worlds. Although bearing the general idea of trading-off between generative and discriminative classifiers [21–23,98,107,108], none of the previous hybrid algorithms demonstrate their utility in challenging segmentation tasks.

**Anomaly Segmentation.** Anomaly segmentation strives to identify unknown object regions, typically in road-driving scenarios [56]. Existing solutions can be generally categorized into three classes: **i)** *Uncertainty estimation* based algorithms [17,18,41–46] usually approximate the uncertainty from simple statistics of the classification probability or logits of pre-trained segmentation models [17,18, 41–43], or adopt Bayesian neural networks with Monte-Carlo dropout to capture pixel uncertainty [44–46]. **ii)** *Outlier exposure* based algorithms make use of auxiliary datasets as training samples of unexpected objects [38–40]. Therefore, this type of algorithms requires re-training the segmentation network, resulting in performance degradation. **iii)** *Image resynthesis* based algorithms reconstruct the input image and discriminate the anomaly instances according to the reconstruction error [34–37].

With a generative classifier, our GMMSeg handles anomaly segmentation naturally, without neither external datasets of outliers, nor additional image resynthesis models. It also greatly differs from most uncertainty estimation-based methods that are post-processing techniques adjusting the prediction scores of softmax-based segmentation networks [17,18,41–43]. The most relevant ones are maybe a few density estimation-based models [56,109,110], which directly measure the likelihood of samples w.r.t.

the data distribution. However, they are either limited to pre-trained representation [56] or specialized for anomaly detection with simple data [109, 110]. To our best knowledge, this is the first time to report promising results on both closed-set and open-world large-scale settings, through a single model instance without any change of network architecture as well as training and inference protocols.

# 3 Methodology

In this section, we first formalize modern semantic segmentation models within a dense discriminative classification framework and discuss defects of such discriminative regime from a probabilistic viewpoint (§3.1). Then we describe our new segmentation framework – GMMSeg – that brings a paradigm shift from the discriminative to generative (§3.2). Finally, in §3.3, we provide implementation details.

## 3.1 Existing Segmentation Solutions: Dense Discriminative Classifier

In the standard semantic segmentation setting, we are given a training dataset $\mathcal{D} = \{(x_n, c_n)\}_{n=1}^N$ of $N$ pairs of pixel samples $x_n \in \mathbb{R}^3$ and corresponding semantic labels $c_n \in \{1, \cdots, C\}$. The goal is to use $\mathcal{D}$ to learn a classification rule which can predict the label $c' \in \{1, \cdots, C\}$ of an unseen pixel $x'$.

Recent mainstream solutions employ a deep neural network for pixel representation learning and softmax for semantic label prediction. Hence they are usually built as a composition of $f \circ g$:
- A *dense feature extractor* $f_{\boldsymbol{\theta}}: \mathbb{R}^3 \rightarrow \mathbb{R}^D$, which is typically an encoder-decoder network that maps the input pixel $x$ to a $D$-dimensional feature representation $\boldsymbol{x}$, *i.e.*, $\boldsymbol{x} = f_{\boldsymbol{\theta}}(x) \in \mathbb{R}^D$;[1] and
- A *dense classifier* $g_{\boldsymbol{\omega}}: \mathbb{R}^D \rightarrow \mathbb{R}^C$, which is achieved by parametric softmax that maps each pixel representation $\boldsymbol{x} \in \mathbb{R}^D$ to $C$ real-valued numbers $\{y_c \in \mathbb{R}\}_{c=1}^C$ termed as *logits*, *i.e.*, $\{y_c\}_{c=1}^C = g_{\boldsymbol{\omega}}(\boldsymbol{x})$, and uses the logits to compute the posterior probability:

$$p(c|\boldsymbol{x}; \boldsymbol{\omega}, \boldsymbol{\theta}) = \frac{\exp(y_c)}{\sum_{c'} \exp(y_{c'})} = \frac{\exp(\boldsymbol{w}_c^\top \boldsymbol{x} + b_c)}{\sum_{c'} \exp(\boldsymbol{w}_{c'}^\top \boldsymbol{x} + b_{c'})} = \frac{\exp(\boldsymbol{w}_c^\top f_{\boldsymbol{\theta}}(x) + b_c)}{\sum_{c'} \exp(\boldsymbol{w}_{c'}^\top f_{\boldsymbol{\theta}}(x) + b_{c'})}, \tag{1}$$

where $\boldsymbol{w}_c \in \mathbb{R}^D$ and $b_c \in \mathbb{R}$ are the weight and bias for class $c$, respectively; and $\boldsymbol{\omega} = \{\boldsymbol{w}_{1:C}, b_{1:C}\}$. The final prediction is the class with the highest predicted probability: $\arg\max_c p(c|\boldsymbol{x}; \boldsymbol{\omega}, \boldsymbol{\theta})$.

The feature extractor $f$ and softmax-based classifier $g$ are jointly trained end-to-end. Their corresponding parameters $\{\boldsymbol{\theta}, \boldsymbol{\omega}\}$ are optimized by minimizing the so-called *cross-entropy* loss on $\mathcal{D}$:

$$\boldsymbol{\theta}^*, \boldsymbol{\omega}^* = \arg\min_{\boldsymbol{\theta}, \boldsymbol{\omega}} -\sum_{(x,c) \in \mathcal{D}} \log p(c|\boldsymbol{x}; \boldsymbol{\omega}, \boldsymbol{\theta}), \tag{2}$$

which is equivalent to maximizing conditional likelihood, *i.e.*, $\Pi_{(x,c) \in \mathcal{D}} p(c|\boldsymbol{x})$. In some literature [11, 111], such learning strategy is called *discriminative training*. As softmax directly models the conditional probability distribution $p(c|\boldsymbol{x})$ with no concern for modeling the input distribution $p(\boldsymbol{x}, c)$, existing softmax-based segmentation models are in essence a dense *discriminative* classifier.

Discriminative softmax typically gives good predictive performance, as the pixel classification rule depends only on the conditional distribution $p(c|\boldsymbol{x})$ in the sense of minimum error rate and softmax optimizes the quantity of interest in a *concise* manner, *i.e.*, learning a direct map from inputs $x$ to the class labels $c$. In spite of its prevalence and effectiveness, this dense discriminative regime has some drawbacks that are still poorly understood: **First**, it attends only to learning the decision boundaries between the $C$ classes on the pixel embedding space, *i.e.*, splitting the $D$-dimensional feature space using $C$ different $(D-1)$-dimensional hyperplanes. It achieves a simplified approach that eliminates extra parameters for modeling the data (representation) distribution [112]. However, from another perspective, it fails to capture the intrinsic class characteristics and is hard to achieve good generalization on unseen data. **Second**, in softmax, each class $c$ corresponds to only a single weight $(\boldsymbol{w}_c, b_c)$. That means existing segmentation models rely on an implicit assumption of *unimodality* of data of each class in the feature space [12, 113, 114]. However, this unimodality assumption is rarely the case in real-world scenarios and makes the model less tolerant of intra-class variances [13], especially when the multimodality remains in the feature space [12]. **Third**, softmax is not capable of inferring the data distribution – it is notorious with inflating the probability of the predicted class as a result of the exponent employed on the network outputs [115]. Thus the prediction score of a class is useless besides its comparative value against other classes. This is the root cause of why existing segmentation models

---

[1]Strictly speaking, the dense feature extractor $f_{\boldsymbol{\theta}}$ typically maps pixel samples with image context, *i.e.*, $f_{\boldsymbol{\theta}}: \mathbb{R}^{h \times w \times 3} \rightarrow \mathbb{R}^{h' \times w' \times D}$, where $h$ and $w$ ($h'$ and $w'$) denote the spatial resolution of the image (feature map) . Here we simplify the notations, *i.e.*, $f_{\boldsymbol{\theta}}: \mathbb{R}^3 \rightarrow \mathbb{R}^D$, to keep a straightforward formulation.

are hard to identify pixel samples $x'$ of an unseen class (out-of-distribution data), $i.e.$, $c' \notin \{1, \cdots, C\}$.

Accordingly, we argue that the time might be right to rethink the current *de facto*, discriminative segmentation regime, where the softmax classifier may actually cause more harm than good.

## 3.2 GMMSeg: Dense GMM Generative Classification

Our GMMSeg reformulates the task from a dense generative classification point of view. Instead of building posterior $p(c|\boldsymbol{x})$ directly, generative classifiers predict labels using Bayes rule. Specifically, generative classifiers model the joint distribution $p(\boldsymbol{x}, c)$, by estimating the class-conditional distribution $p(\boldsymbol{x}|c)$ along with the class prior $p(c)$. Then, following Bayes rule, the posterior is derived as:

$$p(c|\boldsymbol{x}) = \frac{p(c)p(\boldsymbol{x}|c)}{\sum_{c'} p(c')p(\boldsymbol{x}|c')}. \tag{3}$$

Since the class probabilities $p(c)$ are typically set as a *uniform* prior (also in our case), estimating the class-conditional distributions ($i.e.$, data densities) $p(\boldsymbol{x}|c)$ is the core and most difficult part of building a generative classifier. It is also worth noting that generative classifiers are optimized by approximating the data distribution $\Pi_{(x,c)\in\mathcal{D}} p(\boldsymbol{x}|c)$, which is called *generative training* [11].

Although discriminative classifiers demonstrate impressive performance in many application tasks, there are several crucial reasons for using generative rather than discriminative classifiers, which can be succinctly articulated by Feynman's mantra "What I cannot create, I do not understand". Surprisingly, generative classifiers have been rarely investigated in modern segmentation models.

Driven by the belief that generative classifiers are the right way to remove the shortcomings of discriminative approaches, we revisit GMM – one of the most classic generative probabilistic classifiers. We couple the generative EM optimization of GMMs with the discriminative learning of the dense feature extractor $f$ – the most successful part of modern segmentation models, leading to a powerful, principled, and dense generative classification based segmentation framework – GMMSeg (Fig. 2).

Specifically, GMMSeg adopts a weighted mixture of $M$ multivariate Gaussians for modeling the pixel data distribution of each class $c$ in the $D$-dimensional embedding space:

$$p(\boldsymbol{x}|c; \boldsymbol{\phi}_c) = \sum_{m=1}^{M} p(m|c; \boldsymbol{\pi}_c)p(\boldsymbol{x}|c, m; \boldsymbol{\mu}_c, \boldsymbol{\Sigma}_c) = \sum_{m=1}^{M} \pi_{cm} \mathcal{N}(\boldsymbol{x}; \boldsymbol{\mu}_{cm}, \boldsymbol{\Sigma}_{cm}). \tag{4}$$

Here $m|c \sim \text{Multinomial}(\boldsymbol{\pi}_c)$ is the prior probability, $i.e.$, $\sum_m \pi_{cm} = 1$; $\boldsymbol{\mu}_{cm} \in \mathbb{R}^D$ and $\boldsymbol{\Sigma}_{cm} \in \mathbb{R}^{D \times D}$ are the mean vector and covariance matrix for component $m$ in class $c$; and $\boldsymbol{\phi}_c = \{\boldsymbol{\pi}_c, \boldsymbol{\mu}_c, \boldsymbol{\Sigma}_c\}$. The mixture nature allows GMMSeg to accurately approximate the data densities and to be superior over softmax assuming unimodality for each class. Each Gaussian component has an independent covariance structure, enabling a flexible local measure of importance along different feature dimensions.

To find the optimal parameters of the GMM classifier, $i.e.$, $\{\boldsymbol{\phi}_c^*\}_{c=1}^{C}$, a standard approach is EM [116], $i.e.$, maximizing the log likelihood over the feature-label pairs $\{(\boldsymbol{x}_n, c_n)\}_{n=1}^{N}$ in the training dataset $\mathcal{D}$:

$$\boldsymbol{\phi}_c^* = \arg\max_{\boldsymbol{\phi}_c} \sum_{\boldsymbol{x}_n : c_n = c} \log p(\boldsymbol{x}_n|c; \boldsymbol{\phi}_c) = \arg\max_{\boldsymbol{\phi}_c} \sum_{\boldsymbol{x}_n : c_n = c} \log \sum_{m=1}^{M} p(\boldsymbol{x}_n, m|c; \boldsymbol{\phi}_c), \tag{5}$$

EM starts with some initial guess at the maximum likelihood parameters $\boldsymbol{\phi}_c^{(0)}$, and then proceeds to iteratively create successive estimates $\boldsymbol{\phi}_c^{(t)}$ for $t = 1, 2, \cdots$, by repeatedly optimizing a $F$ function [117]:

$$\textbf{E-Step:} \quad q_c^{(t)} = \arg\max_{q_c} F(q_c, \boldsymbol{\phi}_c^{(t-1)}), \qquad \textbf{M-Step:} \quad \boldsymbol{\phi}_c^{(t)} = \arg\max_{\boldsymbol{\phi}_c} F(q_c^{(t)}, \boldsymbol{\phi}_c). \tag{6}$$

$q_c[m] = p(m|\boldsymbol{x}, c; \boldsymbol{\phi}_c)$ gives the probability that data $\boldsymbol{x}$ is *assigned* to component $m$. $F$ is defined as:

$$F(q_c, \boldsymbol{\phi}_c) = \mathbb{E}_{q_c}[\log p(\boldsymbol{x}, m|c; \boldsymbol{\phi}_c)] + H(q_c), \tag{7}$$

where $\mathbb{E}_{q_c}[\cdot]$ gives the expectation w.r.t. the distribution over the $M$ components given by $q_c$, and $H(q_c) = -\mathbb{E}_{q_c}[\log q_c[m]]$ defines the entropy of $q_c$. Based on Eqs. 4-7, for $\forall \boldsymbol{x}_n : c_n = c$, we have:

$$\textbf{E-Step:} \quad q_{cn}^{(t)}[m] = \frac{\pi_{cm}^{(t-1)} \mathcal{N}(\boldsymbol{x}_n | \boldsymbol{\mu}_{cm}^{(t-1)}, \boldsymbol{\Sigma}_{cm}^{(t-1)})}{\sum_{m'=1}^{M} \pi_{cm'}^{(t-1)} \mathcal{N}(\boldsymbol{x}_n | \boldsymbol{\mu}_{cm'}^{(t-1)}, \boldsymbol{\Sigma}_{cm'}^{(t-1)})},$$

$$\textbf{M-Step:} \quad \pi_{cm}^{(t)} = \frac{N_{cm}^{(t)}}{N_c}, \quad \boldsymbol{\mu}_{cm}^{(t)} = \frac{1}{N_{cm}^{(t)}} \sum_{\boldsymbol{x}_n : c_n = c} q_{cn}^{(t)}[m] \boldsymbol{x}_n, \quad \boldsymbol{\Sigma}_{cm}^{(t)} = \frac{1}{N_{cm}^{(t)}} \sum_{\boldsymbol{x}_n : c_n = c} q_{cn}^{(t)}[m](\boldsymbol{x}_n - \boldsymbol{\mu}_{cm}^{(t)})(\boldsymbol{x}_n - \boldsymbol{\mu}_{cm}^{(t)})^\top, \tag{8}$$

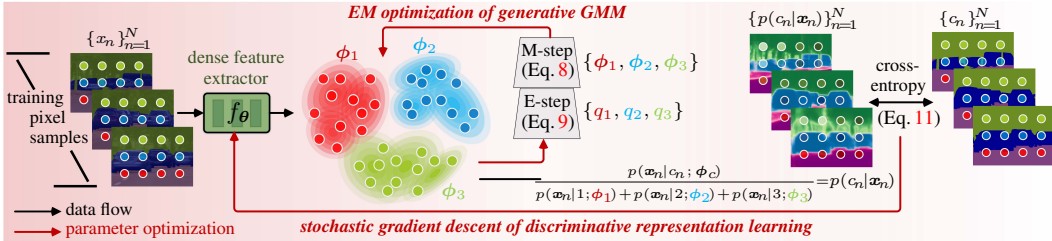

Figure 2: Through generative-discriminative hybrid training, GMMSeg gains the best of the two worlds.

where $N_c$ is the number of training samples labeled as $c$ and $N_{cm} = \sum_{n:c_n=c} q_{cn}[m]$. In E-step, we re-compute the posterior $q_c^{(t)}$ over the $M$ components given the old parameters $\phi^{(t-1)}$. In M-step, with the soft *cluster* assignment $q_c^{(t)}$, the parameters are updated as $\phi_c^{(t)}$ such that the $F$ function is maximized.

In practice, we find standard EM suffers from slow convergence and delivers unsatisfactory results (*cf.* §4.3). A potential reason is the parameter sensitivity of EM – convergent parameters may change vastly even with slightly different initialization [118]. Drawing inspiration from recent optimal transport (OT) based clustering algorithms [119, 120], we introduce a uniform prior on the mixture weights $\boldsymbol{\pi}_c$, *i.e.*, $\forall c, m : \pi_{cm} = \frac{1}{M}$. Recalling $q_c[m] = p(m|\boldsymbol{x}, c)$, we can derive a constraint $\mathcal{Q}_c = \{q_c : \frac{1}{N_c} \sum_{\boldsymbol{x}_n:c_n=c} p(m|\boldsymbol{x}_n, c) = \frac{1}{M}\}$. Then E-step in Eq. 6 is performed by restricting the optimization of $q_c$ over the set $\mathcal{Q}_c$:

$$\textbf{E-Step:} \quad q_c^{(t)} = \arg\max_{q_c \in \mathcal{Q}_c} F(q_c, \phi_c^{(t-1)}). \tag{9}$$

This can be intuitively viewed as an *equipartition* constraint guided clustering process: inside each class $c$, we expect the $N_c$ pixel samples to be evenly *assigned* to $M$ components. As indicated by [28], Eq. 9 is analogous to entropy-regularized OT:

$$\min_{\boldsymbol{Q}_c \in \mathcal{Q}'_c} \sum_{n,m} \boldsymbol{Q}_c(n,m) \boldsymbol{O}_c(n,m) + \epsilon H(\boldsymbol{Q}_c), \quad \mathcal{Q}'_c = \{\boldsymbol{Q}_c \in \mathbb{R}_+^{N_c \times M} : \boldsymbol{Q}_c \mathbf{1}^M = \mathbf{1}^{N_c}, (\boldsymbol{Q}_c)^\top \mathbf{1}^{N_c} = \frac{N_c}{M} \mathbf{1}^M\}, \tag{10}$$

where the transport matrix $\boldsymbol{Q}_c$ (*i.e.*, target solution) can be viewed as the posterior distribution $q_c$ of $N_c$ samples over the $M$ components (*i.e.*, $\boldsymbol{Q}_c(n,m) = q_{cn}[m]$), the cost matrix $\boldsymbol{O}_c \in \mathbb{R}^{N_c \times M}$ is given as the negative log-likelihood, *i.e.*, $\boldsymbol{O}_c(n,m) = -\log p(\boldsymbol{x}_n|c,m)$, and the entropy $H(\cdot)$ is penalized by $\epsilon$. The set $\mathcal{Q}'_c$ encapsulates all the desired constraints over $\boldsymbol{Q}_c$, where $\mathbf{1}^M$ is a $M$-dimensional all-ones vector. Intuitively, the more plausible a pixel sample $\boldsymbol{x}_n$ is with respect to component $m$, the less it costs to transport the underlying mass. Eq. 10 can be efficiently solved via Sinkhorn-Knopp Iteration [120], where $\epsilon$ is set as the default (*i.e.*, 0.05). This optimization scheme, called *Sinkhorn EM*, is proved to have the same global optimum with the EM in Eq. 9 yet is less prone to getting stuck in local optima [28], which is in line with our empirical results (*cf.* §4.3).

Our GMMSeg adopts a hybrid training strategy that is partly generative and partly discriminative:

**Generative Optimization** (Sinkhorn EM) of **GMM Classifier**: $\{\phi_c^*\}_{c=1}^C =$

$$\{\arg\max_{\phi_c} \sum_{\boldsymbol{x}_n:c_n=c} \log p(\boldsymbol{x}_n|c; \phi_c)\}_{c=1}^C = \{\arg\max_{\phi_c} \sum_{\boldsymbol{x}_n:c_n=c} \log \sum_{m=1}^M \pi_{cm} \mathcal{N}(\boldsymbol{x}_n; \boldsymbol{\mu}_{cm}, \boldsymbol{\Sigma}_{cm})\}_{c=1}^C,$$

**Discriminative Learning** (Cross-Entropy Loss) of **Dense Representation**: $\boldsymbol{\theta}^* =$ $\qquad\qquad(11)$

$$\arg\min_{\boldsymbol{\theta}} -\sum_{(x,c) \in \mathcal{D}} \log p(c|\boldsymbol{x}; \{\phi_c^*\}_{c=1}^C, \boldsymbol{\theta}) = \arg\min_{\boldsymbol{\theta}} -\sum_{(x,c) \in \mathcal{D}} \log \Big( \frac{\sum_{m=1}^M \pi_{cm} \mathcal{N}(f_{\boldsymbol{\theta}}(x); \boldsymbol{\mu}_{cm}, \boldsymbol{\Sigma}_{cm})}{\sum_{c'=1}^C \sum_{m=1}^M \pi_{c'm} \mathcal{N}(f_{\boldsymbol{\theta}}(x); \boldsymbol{\mu}_{c'm}, \boldsymbol{\Sigma}_{c'm})} \Big).$$

In GMMSeg, GMM classifier (has $C \times M$ components in total) is purely optimized in a *generative* fashion, *i.e.*, applying Sinkhorn EM to model the data densities $p(\boldsymbol{x}|c)$ within each class $c$ in the feature space $f_{\boldsymbol{\theta}}$. The feature extractor/space $f_{\boldsymbol{\theta}}$, in contrast, is end-to-end trained in a *discriminative* manner, *i.e.*, minimizing the cross-entropy loss over the posteriors output by the GMM. During each training iteration, the extractor's parameters $\boldsymbol{\theta}$ are *only* updated by the gradient backpropagated from the discriminative loss, while the GMM's parameters $\{\phi_c\}_c$ are *only* optimized by EM. To accurately estimate the GMM distributions, an external memory is adopted to store a large set of pixel representations, sampled from several preceding training batches, enabling large-scale EM. Moreover, since the feature space $f_{\boldsymbol{\theta}}$ gradually evolves during training, we opt for a momentum EM: we directly use the GMM's parameters $\{\hat{\phi}_c\}_c$ estimated in the latest iteration as the initial guess in the current

iteration $\{\phi_c^{(0)}\}_c$, and adopt *momentum* update in the M-Step, *i.e.*, $\{\phi_c^{(t)} \leftarrow (1-\tau)\phi_c^{(t)} + \tau\hat{\phi}_c\}_c$, where the momentum coefficient is set as $\tau = 0.999$. This makes our training more stable and accelerates the convergence of EM – we empirically find even one EM loop per training iteration is good enough.

This hybrid training scheme brings several advantages: **First**, GMMSeg achieves the merits of both generative and discriminative learning. The *online* EM based generative optimization enables the GMM to best fit the data distribution even on the evolving feature space. On the other hand, the feature space is discriminatively end-to-end trained under the guidance of the GMM classifier, so as to maximize the pixel-wise predictive performance. **Second**, as the generative EM optimization and discriminative stochastic training work in an independent yet closely collaborative manner, GMMSeg is fully compatible with modern segmentation network architectures and existing discriminative training objectives. It can be further advanced with the development of network architectures of the discriminative counterparts. **Third**, as GMMSeg explicitly models class-conditional data distribution $p(\boldsymbol{x}|c)$, it can naturally handle off-manifold examples, *i.e.*, directly giving meaningful likelihood of the example fitting each class GMM distribution (see §4.2 for experiments on anomaly segmentation).

### 3.3 Implementation Details

**Network Architecture.** GMMSeg is a general framework that can be built upon any modern segmentation network by replacing softmax with the GMM classifier. In our experiments (*cf.* §4.1), we approach GMMSeg on a variety of segmentation models [7, 47–49] and backbones [50, 51]. In the GMM classifier, a $1\times1$ conv is used to compress each pixel feature to a 64-dimensional vector, *i.e.*, $D = 64$, and the covariance matrices $\boldsymbol{\Sigma} \in \mathbb{R}^{D\times D}$ are constrained to be diagonal, for computational efficiency. In our implementation, each class $c$ is represented by a mixture of $M = 5$ Gaussians (there are a total of $5C$ Gaussian components for a segmentation task with $C$ semantic classes). Furthermore, we adopt the *winner-take-all* assumption [121, 122], *i.e.*, the class-wise responsibility (Eq. 4) is dominated by the largest term, for better performance.

**Training** In each training iteration, we conduct one loop of momentum (Sinkhorn) EM (*i.e.*, $t=1$) on current training batch as well as the external memory for the generative optimization of GMM, and backpropagate the gradient of the cross-entropy loss on current batch for the discriminative training of the feature extractor. The external memory maintains a queue for each component in each class; each queue gathers 32K pixel features from previous training batches in a *first in, first out* manner. To improve the diversity of the stored pixel features, we sample a sparse set of 100 pixels per class from each image, instead of directly storing the whole images into the memory. Note that the memory is discarded after training, and does not introduce extra overheads in inference.

**Inference.** GMMSeg only brings negligible delay in the inference speed compared to the discriminative counterparts (see experiments in §4.3). For standard (closed-set) semantic segmentation, pixel prediction is made using Bayes rule (*cf.* Eq. 3): $\arg\max_c p(c|\boldsymbol{x})$, where $p(c|\boldsymbol{x}) \propto p(\boldsymbol{x}|c)$ with the uniform class distribution prior: $p(c) = 1/C$. For anomaly segmentation, the pixel-wise uncertainty/anomaly score can be naturally raised as: $-\max_c p(\boldsymbol{x}|c)$, *i.e.*, the outlier input should reside in low-probability regions [123].

## 4 Experiments

We respectively examine the efficacy and robustness of GMMSeg on semantic segmentation (§4.1) and anomaly segmentation (§4.2). In §4.3, we provide diagnostic analysis on our core model design.

### 4.1 Experiments on Semantic Segmentation

**Datasets.** We conduct experiments on three widely used semantic segmentation datasets:
- ADE$_{20K}$ [53] has 20K/2K/3K images in `train/val/test` set, with 150 stuff/object categories in total.
- Cityscapes [54] has 2,975/500/1,524 fine-labeled images for `train/val/test` set with 19 classes.
- COCO-Stuff [55] has 10K images (9K/1K for `train/test`), pixel-wise labeled with 171 classes.

**Base Segmentation Architectures and Backbones.** For thorough evaluation, we apply GMMSeg to four famous segmentation architectures (*i.e.*, DeepLab$_{V3+}$[47], OCRNet [48], UPerNet [49], Segformer [7]), with various backbones (*i.e.*, ResNet [50], HRNet [51], Swin [52], MiT [7]). For fairness, we re-implement these models using the standardized hyper-parameter setting in MMSegmentation [124].

**Training Details.** GMMSeg is implemented on MMSegmentation [124] and follows the standard training setting for each dataset. All models are initialized with ImageNet-1K [125] pretrained back-

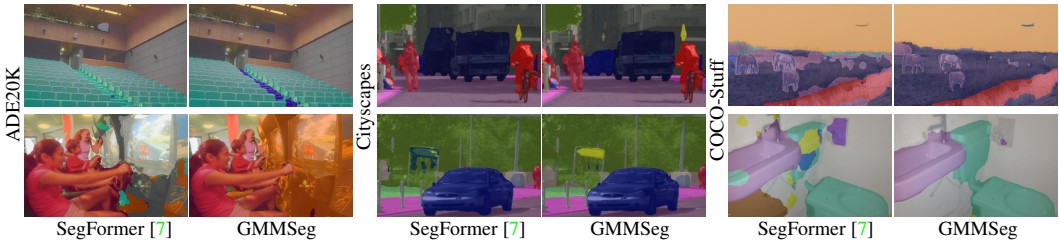

Figure 3: Qualitative results (§4.1) on ADE$_{20K}$ [53], Cityscapes [54], and COCO-Stuff [55].

SegFormer [7]  GMMSeg  SegFormer [7]  GMMSeg  SegFormer [7]  GMMSeg

bones and trained with commonly used data augmentations including resizing, flipping, color jittering and cropping. For ADE$_{20K}$/COCO-Stuff/Cityscapes, images are cropped to $512\times512/512\times512/768\times768$ and models are trained for 160K/80K/80K iterations with 16/16/8 batch size, using 8/16 NVIDIA Tesla A100 GPUs. Other training hyper-parameters (*i.e.*, optimizers, learning rates, weight decays, schedulers) are set as the default in MMSegmentation and can be found in the supplementary.

**Inference Details.** For ADE$_{20K}$ and COCO-Stuff, we keep the aspect ratio of test images and rescale the short side to 512. For Cityscapes, sliding window inference is used with $768\times768$ window size. Note that for fairness, all our results are reported without any test-time data augmentation.

**Quantitative Results.** Table 1 demonstrates our quantitative results. Although mainly focusing on the comparison with the four base segmentation models [7, 47–49], we further include five widely recognized methods [1, 3, 8, 9, 90] for completeness. As can be seen, our GMMSeg outperforms all its discriminative counterparts across various datasets, backbones, and network architectures (FCN-style and Transformer-like):

- ADE$_{20K}$ [53] `val`. With FCN-style segmentation neural architectures, *i.e.*, DeepLab$_{V3+}$ and OCR, GMMSeg provides **1.2**%/**1.5**% mIoU gains over corresponding discriminative models. Similar performance improvements, *i.e.*, **1.0**% and **0.6**%, are also obtained with attentive neural architectures, *i.e.*, Swin-UperNet and SegFormer, manifesting the universality and efficacy of GMMSeg.
- Cityscapes [54] `val`. Again our GMMSeg surpasses all its discriminative counterparts by large margins, *e.g.*, **0.5**% over DeepLab$_{V3+}$, **0.8**% over OCRNet, **0.7**% over Swin-UperNet, and **0.6**% over SegFormer, suggesting its wide utility in this field.
- COCO-Stuff [55] `test`. Our GMMSeg also demonstrates promising results. This is particularly impressive considering these results are achieved by a dense generative classifier, while the semantic segmentation task is commonly considered as a battlefield for discriminative approaches.

Table 1: Quantitative results (§4.1) on ADE$_{20K}$ [53] `val`, Cityscapes [54] `val`, and COCO-Stuff [55] `test` with mean IoU.

| Method | Backbone | ADE$_{20K}$ | Citys. | COCO. |
|---|---|---|---|---|
| FCN [CVPR15] [1] | ResNet$_{101}$ | 39.9 | 75.5 | 32.6 |
| PSPNet [CVPR17] [3] | ResNet$_{101}$ | 44.4 | 79.8 | 37.8 |
| SETR [CVPR21] [9] | †ViT$_{Large}$ | 48.2 | 79.2 | - |
| Segmenter [ICCV21] [8] | †ViT$_{Large}$ | ‡51.8 | 79.1 | - |
| MaskFormer [NeurIPS21] [90] | †Swin$_{Base}$ | ‡52.7 | - | - |
| DeepLab$_{V3+}$ [ECCV18] [47] | ResNet$_{101}$ | 45.5 | 80.6 | 33.8 |
| GMMSeg | | 46.7↑1.2 | 81.1↑0.5 | 35.5↑1.7 |
| OCRNet [ECCV20] [48] | HRNet$_{V2W48}$ | 43.3 | 80.4 | 37.6 |
| GMMSeg | | 44.8↑1.5 | 81.2↑0.8 | 39.2↑1.6 |
| UPerNet [ECCV18] [49] | Swin$_{Base}$ | 48.0 | 81.1 | 43.4 |
| GMMSeg | | 49.0↑1.0 | 81.8↑0.7 | 44.3↑0.9 |
| SegFormer [NeurIPS21] [7] | MiT$_{B5}$ | 50.0 | 82.0 | 44.0 |
| GMMSeg | | 50.6↑0.6 | 82.6↑0.6 | 44.7↑0.7 |

†: pretrained on ImageNet$_{22K}$; ‡: using larger crop-size, *i.e.*, $640\times640$

**Qualitative Results.** In Fig. 3, we illustrate the qualitative comparisons of our GMMSeg against SegFormer [7]. It is evident that, among the representative samples in the three datasets, our method yields more accurate predictions when facing challenging scenarios, *e.g.*, unconspicuous objects.

## 4.2  Experiments on Anomaly Segmentation

**Datasets.** To fully reveal the merits of our generative method, we next test its robustness for abnormal data, *i.e.*, identifying test samples of unseen classes, using two popular anomaly segmentation datasets:
- Fishyscapes Lost&Found [56], built upon [126], has 100/275 `val/test` images. It is collected under the same setup as Cityscapes [54] but with real obstacles on the road. Pixels are labeled as either background (*i.e.*, pre-defined Cityscapes classes) or anomaly (*i.e.*, other unexpected classes like crate).
- Road Anomaly [36] has 60 images containing anomalous objects in unusual road conditions.

**Evaluation Metrics.** The area under receiver operating characteristics (AUROC), average precision (AP), and false positive rate (FPR$_{95}$) at a true positive rate of 95%, are adopted following [18, 35, 56].

Table 2: Quantitative results (§4.2) on Fishyscapes Lost&Found [56] `val` and Road Anomaly [36].

| Method | Extra Resyn. | OOD Data | mIoU | Fishyscapes Lost&Found | | | Road Anomaly | | |
|---|---|---|---|---|---|---|---|---|---|
| | | | | AUROC↑ | AP↑ | FPR$_{95}$↓ | AUROC↑ | AP↑ | FPR$_{95}$↓ |
| SynthCP [ECCV20] [35] | ✓ | ✓ | 80.3 | 88.34 | 6.54 | 45.95 | 76.08 | 24.86 | 64.69 |
| SynBoost [CVPR21] [34] | ✓ | ✓ | - | 96.21 | 60.58 | 31.02 | 81.91 | 38.21 | 64.75 |
| MSP [ICLR17] [17] | ✗ | ✗ | 80.3 | 86.99 | 6.02 | 45.63 | 73.76 | 20.59 | 68.44 |
| Entropy [ICLR17] [17] | ✗ | ✗ | 80.3 | 88.32 | 13.91 | 44.85 | 75.12 | 22.38 | 68.15 |
| SML [ICCV21] [18] | ✗ | ✗ | 80.3 | 96.88 | 36.55 | 14.53 | 81.96 | 25.82 | 49.74 |
| *Mahalanobis [NeurIPS18] [19] | ✗ | ✗ | 80.3 | 92.51 | 27.83 | 30.17 | 76.73 | 22.85 | 59.20 |
| *GMMSeg-DeepLab$_{V3+}$ | ✗ | ✗ | 81.1 | 97.34 | 43.47 | 13.11 | 84.71 | 34.42 | 47.90 |
| *GMMSeg-FCN | ✗ | ✗ | 76.7 | 96.28 | 32.94 | 16.07 | 78.99 | 24.51 | 56.95 |
| *GMMSeg-SegFormer | ✗ | ✗ | 82.6 | 97.83 | 50.03 | 12.55 | 89.37 | 57.65 | 44.34 |

∗: confidence derived with generative formulation

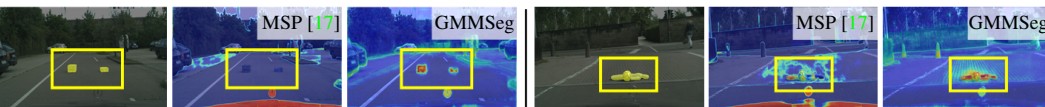

Figure 4: Qualitative results (§4.2) of anomaly heatmaps on Fishyscapes Lost&Found [56] `val`.

**Experiment Protocol.** As in [17, 18, 127], we adopt ResNet$_{101}$-DeepLab$_{V3+}$ architecture. For completeness, we also report the results of our GMMSeg based on ResNet$_{101}$-FCN and MiT$_{B5}$-SegFormer. All our models are the same ones in Table 1, *i.e.*, trained on Cityscapes `train` only. As GMMSeg estimates class densities $p(\boldsymbol{x}|c)$, it can naturally reject unlikely inputs (*cf.* §3.3), *i.e.*, directly thresholding $-\max_c p(\boldsymbol{x}|c)$ for computing the anomaly segmentation metrics, *without any post-processing*.

**Quantitative Results.** As shown in Table 2, based on DeepLab$_{V3+}$ architecture, GMMSeg outperforms all the competitors under the same setting, *i.e.*, neither using external out-of-distribution data nor extra resynthesis module. Note that, [17–19] rely on pre-trained discriminative segmentation models and thus have to make post-calibration. However, GMMSeg directly derives *meaningful* confidence scores from likelihood $p(\boldsymbol{x}|c)$. Mahalanobis [19] also models data density, yet, merely on pre-trained feature space with a single Gaussian per class. In contrast, GMMSeg performs much better, proving the superiority of mixture modeling and hybrid training. Even with a weaker architecture, *i.e.*, FCN, GMMSeg still performs robustly. When adopting SegFormer, better performance is achieved.

**Qualitative Results.** In Fig. 4, we visualize the anomaly score heatmaps generated by MSP [17]-DeepLab$_{V3+}$ [47] and GMMSeg-DeepLab$_{V3+}$. The softmax based counterpart ignores the anomalies with overconfident predictions; in contrast, GMMSeg naturally rejects them (red colored regions).

### 4.3 Diagnostic Experiments

For in-depth analysis, we conduct ablative studies using DeepLab$_{V3+}$[47]-ResNet$_{101}$[50] segmentation architecture. Due to limited space, we put some diagnostic experiments in our supplementary material.

**Online Hybrid Training.** We first investigate our hybrid training strategy (*cf.* Eq. 11), where the discriminative feature extractor and generative GMM classifier are online optimized iteratively. Owe to this ingenious design, both components are gradually updated, aligned with and adaptive to each other, making GMMSeg

Table 3: Online hybrid training (§4.3), evaluated on ADE$_{20K}$ [53].

| Method | mIoU (%) |
|---|---|
| DeepLab$_{V3+}$ + GMM | 31.6 |
| GMMSeg-DeepLab$_{V3+}$ | 46.0 |

a compact model. To fully demonstrate the effectiveness, we study a variant, DeepLab$_{V3+}$+GMM, where a GMM classifier is directly fitted onto the feature space trained with the softmax classifier beforehand. As shown in Table 3, a clear performance drop is observed, *i.e.*, mIoU: $46.0\% \rightarrow 31.6\%$, revealing the appealing efficacy of our end-to-end hybrid training strategy.

**Discriminative GMMSeg *vs.* Generative GMMSeg.** Our GMMSeg learns generative GMM via EM, *i.e.*, $\max p(\boldsymbol{x}|c; \boldsymbol{\phi})$, with discri-

Table 4: Discriminative GMMSeg *vs.* generative GMMSeg (§4.3).

| GMMSeg | Training Objective | Cityscapes | Fishyscapes Lost&Found | | |
|---|---|---|---|---|---|
| | | mIoU↑ | AUROC↑ | AP↑ | FPR$_{95}$↓ |
| Discriminative | $\max p(c|\boldsymbol{x}; \boldsymbol{\phi}, \boldsymbol{\theta})$ | 81.0 | 89.77 | 17.68 | 51.81 |
| Generative | $\max p(\boldsymbol{x}|c; \boldsymbol{\phi}) + \max p(c|\boldsymbol{x}; \boldsymbol{\theta})$ | 81.1 | 97.34 | 43.47 | 13.11 |

minative representation learning, *i.e.*, $\max p(c|\boldsymbol{x}; \boldsymbol{\theta})$. A discriminative counterpart can be achieved by end-to-end learning all the parameters, *i.e.*, $\{\boldsymbol{\phi}, \boldsymbol{\theta}\}$, with cross-entropy loss, *i.e.*, $\max p(c|\boldsymbol{x}; \boldsymbol{\phi}, \boldsymbol{\theta})$. Discriminative GMMSeg sacrifices data characterization for more flexiblility in discrimination, and yields poor performance in open-world setting. While inapparent effect on closed-set Cityscapes is observed, which in turn verifies the accurate specification of data distribution in generative GMMSeg.

**Standard EM *vs*. Sinkhorn EM.** In our GMMSeg, we leverage the entropic OT based Sinkhorn EM [28] (*cf*. Eq. 10) instead of the classic one (*cf*. Eq. 8) for the generative optimization of the GMM. In Table 5a, we investigate the impacts of these two different EM algorithms and show that Sinkhorn EM is more favored. More specifically, during the E-step, rather than the vanilla EM assigning data samples to Gaussian components independently, Sinkhorn EM restricts the assignment with an equipartition constraint. As pointed out in [28], incorporating such prior information about the mixing weights of GMM components leads to higher curvature around the global optimum. Our empirical results confirm this theoretical finding.

Table 5: **Ablative studies** (§4.3) on ADE$_{20K}$ [53] `val`. The adopted settings are marked in red.

| EM algorithm | # Loop | mIoU (%) | # Component | mIoU (%) |
|---|---|---|---|---|
| vanilla EM | 1 | 42.7 | $M = 1$ | 44.2 |
| | 10 | 44.8 | $M = 3$ | 45.3 |
| Sinkhorn EM | 1 | 46.0 | $M = 5$ | 46.0 |
| | 5 | 46.0 | $M = 10$ | 46.0 |
| | 10 | 46.0 | $M = 15$ | 45.7 |

(a) EM optimization      (b) # Component *per* class

**Number of EM Loop per Training Iteration.** EM algorithm alternates between E-step and M-step for maximum-likelihood inference (*cf*. Eq. 6). In GMMSeg, in order to blend EM with stochastic gradient descent, we adopt an online version of (Sinkhorn) EM based on momentum update. In Table 5a, we also study the influence of looping EM different times per training iteration. We can find that one loop per iteration is enough to catch the drift of the gradually updated feature space.

**Number of Gaussian Components per Class.** In GMMSeg, data distribution of each class is modeled by a mixture of $M$ Gaussian components (*cf*. Eq. 4). Table 5b shows the results with different values of $M$. When $M = 1$, each class corresponds to a single Gaussian, which is directly estimated via Gaussian Discriminant Analysis, without EM. This baseline achieves $44.2\%$ mIoU. After adopting the mixture model, *i.e*., $M : 1 \rightarrow 3 \rightarrow 5$, the performance is greatly improved, *i.e*., mIoU: $44.2\% \rightarrow 45.3\% \rightarrow 46.0\%$. This verifies our hypothesis of class multimodality. Yet, further increasing component number (*i.e*., $M : 5 \rightarrow 15$) only brings marginal even negative gains, due to overparameterization.

**Confidence Calibration.** We further study the model calibration of GMMSeg and the discriminative counterpart, *i.e*., DeepLab$_{V3+}$ [49] with the softmax classifier. In Fig. 5, we illustrate the Expected Calibration Error (ECE) [15] along with reliability diagrams, which plot the expected pixel accuracy as a function of confidence [15]. As seen, GMMSeg yields better calibrated predictions, *i.e*., smaller gaps between the expected accuracy and confidence. On the

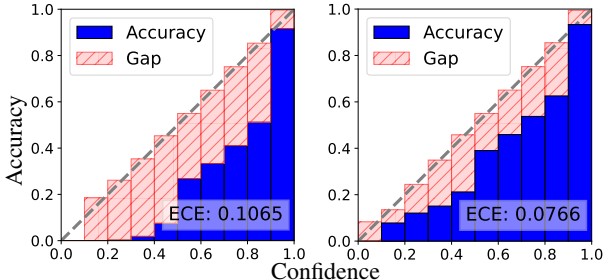

Figure 5: Reliability diagrams for DeepLab$_{V3+}$ [49] (left) and GMMSeg-DeepLab$_{V3+}$ (right) on Cityscapes `val`.

other hand, the discriminative softmax produces confidences that deviate more from the true probabilities, and suffers higher calibration error accordingly. This again verifies the better reliability and interpretability of GMMSeg compared to its discriminative counterparts.

**Runtime Analysis.** The inference speed of GMMSeg is 13.37 fps, which only yields negligible overhead w.r.t. its discriminative softmax counterpart, *i.e*., 13.37 *vs*. 14.16 fps. We measure the fps on a single NVIDIA GeForce RTX 3090 GPU with a batch size of one.

## 5 Conclusion

We presented GMMSeg, the first generative neural framework for semantic segmentation. By explicitly modeling data distribution as GMMs, GMMSeg shows promise to solve the intrinsic limitations of current softmax based discriminative regime. It successfully optimizes generative GMM with end-to-end discriminative representation learning in a compact and collaborative manner. This makes GMMSeg principled and well applicable in both closed-set and open-world settings. We believe this work provides fundamental insights and can benefit a broad range of application tasks. As a part of our future work, we will explore our algorithm in image classification and trustworthy AI related tasks.

**Acknowledgement.** This work was partially supported by the Fundamental Research Funds for the Central Universities (No. 226-2022-00087), and by the National Key R&D Program of China (No. 2020AAA0108800). Wenguan Wang acknowledges partial support from Australian Research Council (ARC), DECRA DE220101390.

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
