# GMMSeg: Gaussian Mixture Models for Deep Generative Semantic Segmentation
## *Supplemental Material*

**Chen Liang**[1,3*†], **Wenguan Wang**[2*], **Jiaxu Miao**[1], **Yi Yang**[1]

[1]CCAI, Zhejiang University    [2]ReLER, AAII, University of Technology Sydney    [3]Baidu Research

https://github.com/leonnnop/GMMSeg

In this document, we provide the following items that shed deeper insight on our contributions:

- §S1: Detailed training parameters.
- §S2: More experimental results.
- §S3: More qualitative visualization.
- §S4: Discussion of legal/ethical considerations and limitations.

## S1   Detailed Training Parameters

We evaluate our GMMSeg on six base segmentation architectures. Four of them, *i.e.*, DeepLab$_{v3+}$ [1], OCRNet [2], Swin-UperNet [3], SegFormer [4], are presented in our main paper. And the two additional base architectures, *i.e.*, FCN [5] and Mask2Former [6], are provided in this supplemental material (*cf*. §S2). We follow the default training settings in the official Mask2Former codebase and MMSegmentation for Mask2Former and other base architectures respectively. In particular, we train FCN, DeepLab$_{v3+}$ and OCRNet using SGD optimizer with initial learning rate 0.1, weight decay 4e-4 with polynomial learning rate annealing; we train Swin-UperNet and SegFormer using AdamW optimizer with initial learning rate 6e-5, weight decay 1e-2 with polynomial learning rate annealing; we train Mask2Former using AdamW optimizer with initial learning rate 1e-4, weight decay 5e-2 and the learning rate is decayed by a factor of 10 at 0.9 and 0.95 fractions of the total training steps.

## S2   More Experimental Results

**More Base Segmentation Architectures.** We first demonstrate the efficacy of our GMMSeg on two additional base segmentation architectures, *i.e.*, FCN [5] and Mask2Former [6], with quantitative results summarized in Table S1. We train FCN based models with the according training hyperparameter settings mentioned in §S1 and strictly follow the same training and inference setups in our main manuscript (*cf*. §4.1). Furthermore, for a fair comparison with Mask2Former, the backbone, *i.e.*, Swin$_{Large}$ [3], is pretrained with ImageNet$_{22K}$ [10].

Table S1: Additional quantitative results (§S2) on ADE$_{20K}$ [7] `val`, Cityscapes [8] `val`, and COCO-Stuff [9] `test` in mean IoU.

| Method | Backbone | ADE$_{20K}$ | Citys. | COCO. |
|---|---|---|---|---|
| FCN [CVPR15] [5] | ResNet$_{101}$ | 39.9 | 75.5 | 32.6 |
| GMMSeg | | **41.8** ↑1.9 | **76.7** ↑1.2 | **34.1** ↑1.5 |
| Mask2Former [CVPR22] [6] | Swin$_{Large}$ | 56.1 | 83.3 | 51.0 |
| GMMSeg | | **56.7** ↑0.6 | **83.8** ↑0.5 | **52.0** ↑1.0 |

For ADE$_{20K}$/COCO-Stuff/Cityscapes, we train Mask2Former based models using images cropped to $640 \times 640$/$640 \times 640$/$1024 \times 1024$, for 160K/80K/90K iterations with 16/16/16 batch size. We adopt sliding window inference on Cityscapes with a window size of $1024 \times 1024$ and we keep the aspect ratio of test images and rescale the short side to 640 on ADE$_{20K}$ and COCO-Stuff.

---

[*]Equal contributions.

[†]Work partly done during an internship at Baidu Research.

Table S2: Quantitative results (§S2) on Fishyscapes (FS) Lost&Found `test` and Static `test`.

| Method | Re-training | Extra Network | OoD Data | FS Lost&Found | | FS Static | |
|---|---|---|---|---|---|---|---|
| | | | | AP ↑ | FPR$_{95}$ ↓ | AP ↑ | FPR$_{95}$ ↓ |
| Density - Single-layer NLL [12] | ✗ | ✓ | ✗ | 3.01 | 32.9 | 40.86 | 21.29 |
| Density - Minimum NLL [12] | ✗ | ✓ | ✗ | 4.25 | 47.15 | 62.14 | 17.43 |
| Density - Logistic Regression [12] | ✗ | ✓ | ✓ | 4.65 | 24.36 | 57.16 | 13.39 |
| Image Resynthesis [15] | ✗ | ✓ | ✗ | 5.70 | 48.05 | 29.6 | 27.13 |
| Bayesian Deeplab [16] | ✓ | ✗ | ✗ | 9.81 | 38.46 | 48.70 | 15.05 |
| OoD Training - Void Class [17] | ✓ | ✗ | ✓ | 10.29 | 22.11 | 45.00 | 19.40 |
| Discriminative Outlier Detection Head [18] | ✓ | ✓ | ✓ | 31.31 | 19.02 | 96.76 | 0.29 |
| Dirichlet Deeplab [19] | ✓ | ✗ | ✓ | 34.28 | 47.43 | 31.30 | 84.60 |
| SynBoost [20] | ✗ | ✓ | ✓ | 43.22 | 15.79 | 72.59 | 18.75 |
| MSP [21] | ✗ | ✗ | ✗ | 1.77 | 44.85 | 12.88 | 39.83 |
| Entropy [22] | ✗ | ✗ | ✗ | 2.93 | 44.83 | 15.41 | 39.75 |
| kNN Embedding - density [12] | ✗ | ✗ | ✗ | 3.55 | 30.02 | 44.03 | 20.25 |
| SML [14] | ✗ | ✗ | ✗ | 31.05 | 21.52 | 53.11 | 19.64 |
| GMMSeg-DeepLab$_{V3+}$ | ✗ | ✗ | ✗ | **55.63** | **6.61** | **76.02** | **15.96** |

Here FCN is a famous fully convolutional model that is in line with the per-pixel dense classification models we discussed in the main paper (*cf*. §3.1). Besides, of particular interest is the Mask2Former, which is an attentive model proposed very recently that formulates the task as a mask classification problem, where a mask-level representation is learned instead of pixel-level. However, it still relies on a discriminative softmax based classifier for mask classification. We equip Mask2Former by replacing the softmax classification module with our generative GMM classifier.

As seen, GMMSeg consistently boosts the model performance despite different segmentation formulations, *i.e.*, pixel classification or mask classification, verifying the superiority of our GMMSeg that brings a paradigm shift from a discriminative softmax to a generative GMM. Notably, with Mask2Former-Swin$_{Large}$ as base segmentation architecture, our GMMSeg earns mIoU scores of **56.7%/83.8%/52.0%**, establishing new state-of-the-arts among ADE$_{20K}$/Cityscapes/COCO-Stuff.

**Anomaly Segmentation Result on Fishyscapes Lost&Found `test` and Static `test`.** We additionally report the anomaly segmentation performance of our Cityscapes [8] trained GMMSeg built upon DeepLab$_{V3+}$ [1]-ResNet$_{101}$ [11] on Fishyscapes [12] Lost&Found `test` and Static `test`. Fishyscapes Static is a blending-based dataset built upon backgrounds from Cityscapes and anomalous objects from Pascal VOC [13], that contains 30/1,000 images in `val`/`test` set. The `test` splits of Fishyscapes Lost&Found and Static are privately held by the Fishyscapes organization that contain entirely unknown anomalies to the methods. The results are summarized in Table S2, and are also publicly available in anonymous on the official leaderboard[3]. We categorize the methods by checking whether they require retraining, extra segmentation networks or utilize OoD data, following [12, 14].

As seen, without any add-on post-calibration technique, GMMSeg significantly surpasses the state-of-the-art methods by even larger margins on the challenging `test` set compared to results on `val` set, *i.e.*, **+24.58%/+14.91%** in AP and **+22.91%/+3.68%** in FPR$_{95}$ on Fishyscapes Lost&Found/Static `test`. Notably, GMMSeg even outperforms all other benchmark methods that employ additional training networks/data on Fishyscapes Lost&Found `test`, verifying the strong robustness to unexpected anomalies on-road due to the accurate data density modeling of GMMSeg.

**Impact of Memory Capacity.** In Table S3, we further explore the influence of the memory capacity, *i.e.*, the amount of pixel representations stored for class-wise EM estimation, with DeepLab$_{V3+}$-ResNet$_{101}$ on ADE$_{20K}$ `val` trained for 80K iterations. For the first row, where the memory size is set to 0, the EM is only performed within mini-batches. Not surprisingly, data distribution estimated at such a local scale is far from accurate, leading to inferior results. With enlarged memory capacity, the performance is increased. When the performance reaches saturation, the stored pixel samples are sufficient enough to represent the true data distribution of the whole training set.

Table S3: Impact of memory size, evaluated on ADE$_{20K}$ [7] `val`.

| # Sample | mIoU (%) |
|---|---|
| 0 | 40.3 |
| 8K | 45.1 |
| 16K | 45.4 |
| 32K | 46.0 |
| 48K | 46.0 |

---

[3] https://fishyscapes.com/results

## S3    More Qualitative Visualization

**Semantic Segmentation.**   We illustrate the qualitative comparisons of GMMSeg equipped Seg-Former [4]-MiT$_{B5}$ against the original model on ADE$_{20K}$ [7] (Fig. S1), Cityscapes [8] (Fig. S2) and COCO-Stuff [9] (Fig. S3). It is evident that, benefiting from the accurate data characterization modeling, GMMSeg is less confused by object categories and gives preciser predictions than SegFormer.

**Anomaly Segmentation.**   We then show more qualitative results of MSP [22]-DeepLab$_{V3+}$ [1] and GMMSeg-DeepLab$_{V3+}$ on Fishyscapes Lost&Found `val`. As observed, different from MSP, GMMSeg gets rid of being overwhelmed by overconfident predictions and successfully identifies the anomalies.

## S4    Discussion

**Asset License and Consent.**   We use three semantic segmentation datasets, *i.e.*, ADE$_{20K}$ [7], Cityscapes [8], COCO-Stuff [9], and two anomaly segmentation datasets, *i.e.*, Fishyscapes [12], Road Anomaly [15], that are all publicly and freely available for academic purposes. We implement all models with MMSegmentation [23] and official Mask2Former [6] codebases. ADE$_{20K}$ (`https://groups.csail.mit.edu/vision/datasets/ADE20K/`) is released under a CC BSD-3; Cityscapes (`https://www.cityscapes-dataset.com/`) is released under this License; COCO-Stuff v1.1 (`https://github.com/nightrome/cocostuff`) is released under Flickr Terms of use for images and CC BY 4.0 for annotations; Road Anomaly (`https://www.epfl.ch/labs/cvlab/data/road-anomaly/`) is released under CC BY 4.0; All assets mentioned above release annotations obtained from human experts with agreements. Fishyscapes (`https://fishyscapes.com/`) is released under CC BY 4.0. This dataset is synthesized and re-organized from existing datasets that we are not capable to trace every detail; MMSegmentation codebase (`https://github.com/open-mmlab/mmsegmentation`) is released under Apache-2.0 license. Mask2Former codebase (`https://github.com/facebookresearch/Mask2Former`) is released under MIT license.

**Limitation Analysis.**   One limitation of our approach is that the EM based generative parameter estimation needs extra optimization loops in each training iteration which would reduce the training efficiency in terms of time complexity. However, in practice, we find one EM loop per training iteration is good enough for global model convergence, which only brings a minor computational overhead, *i.e.*, $\sim 5\%$ training speed delay. We will dedicate to designing more powerful algorithms with further improvements in both efficiency and efficacy.

**Broader Impact.** This work introduces the first generative semantic segmentation framework that shows promising results in both closed-set and open-world scenarios. On positive side, the approach advances model accuracy of semantic segmentation and can certainly have a wide range of real-world applications, *e.g.*, precision agriculture, robot navigation, *etc*. The strong robustness to anomalous objects further warrants its potential for usage in safety-critical applications, *i.e.*, autonomous driving. On negative side, the generated results can be fed into other algorithms for malicious purposes, *e.g.*, identifying the minority groups. Though beyond the scope of this paper, we will organize a gated release of our models to make sure that they are not being used beyond academic research purposes.

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

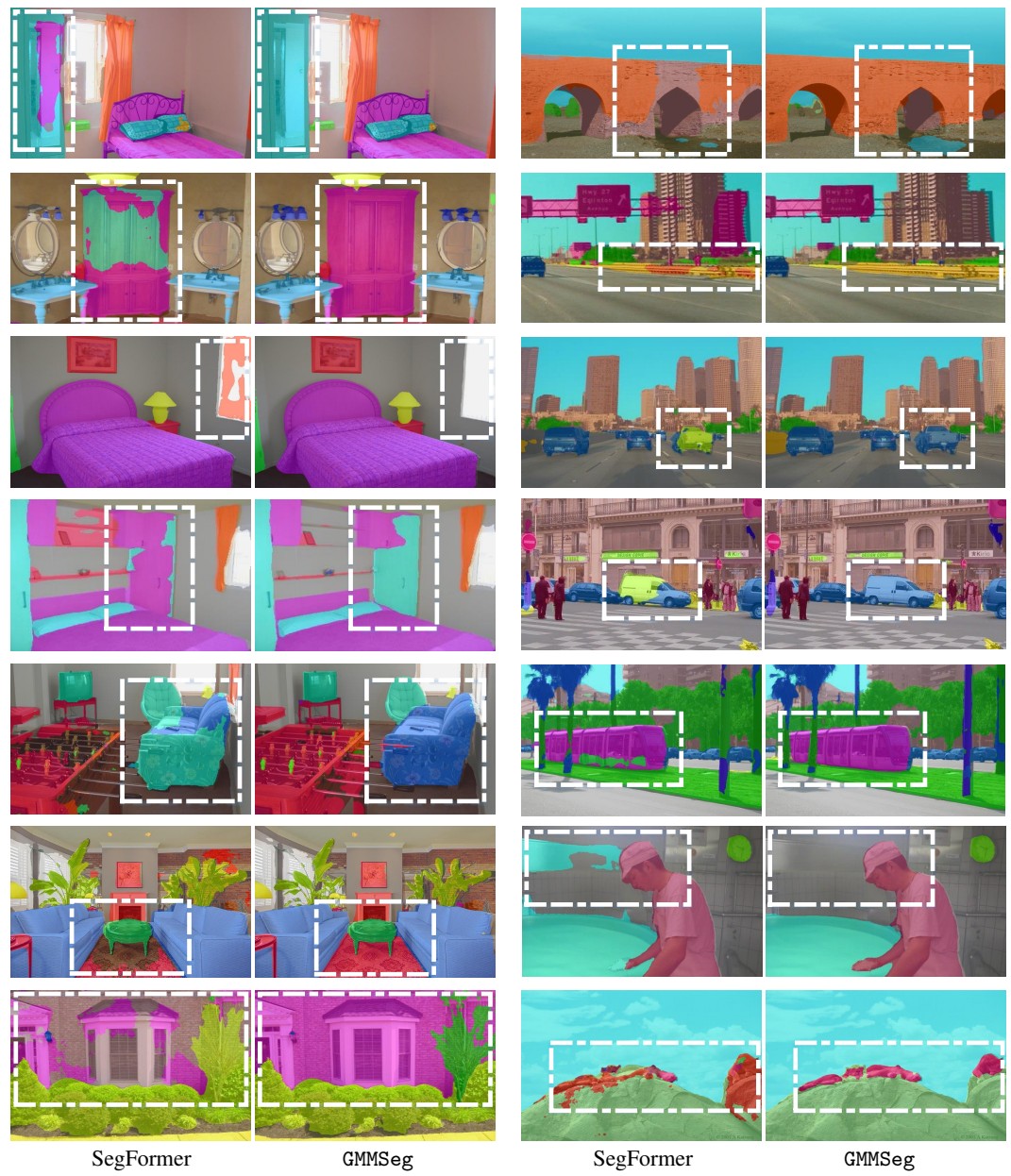

| SegFormer | GMMSeg | SegFormer | GMMSeg |

Figure S1: Qualitative results (§S3) of SegFormer [4] and our GMMSeg on ADE$_{20K}$ [7].

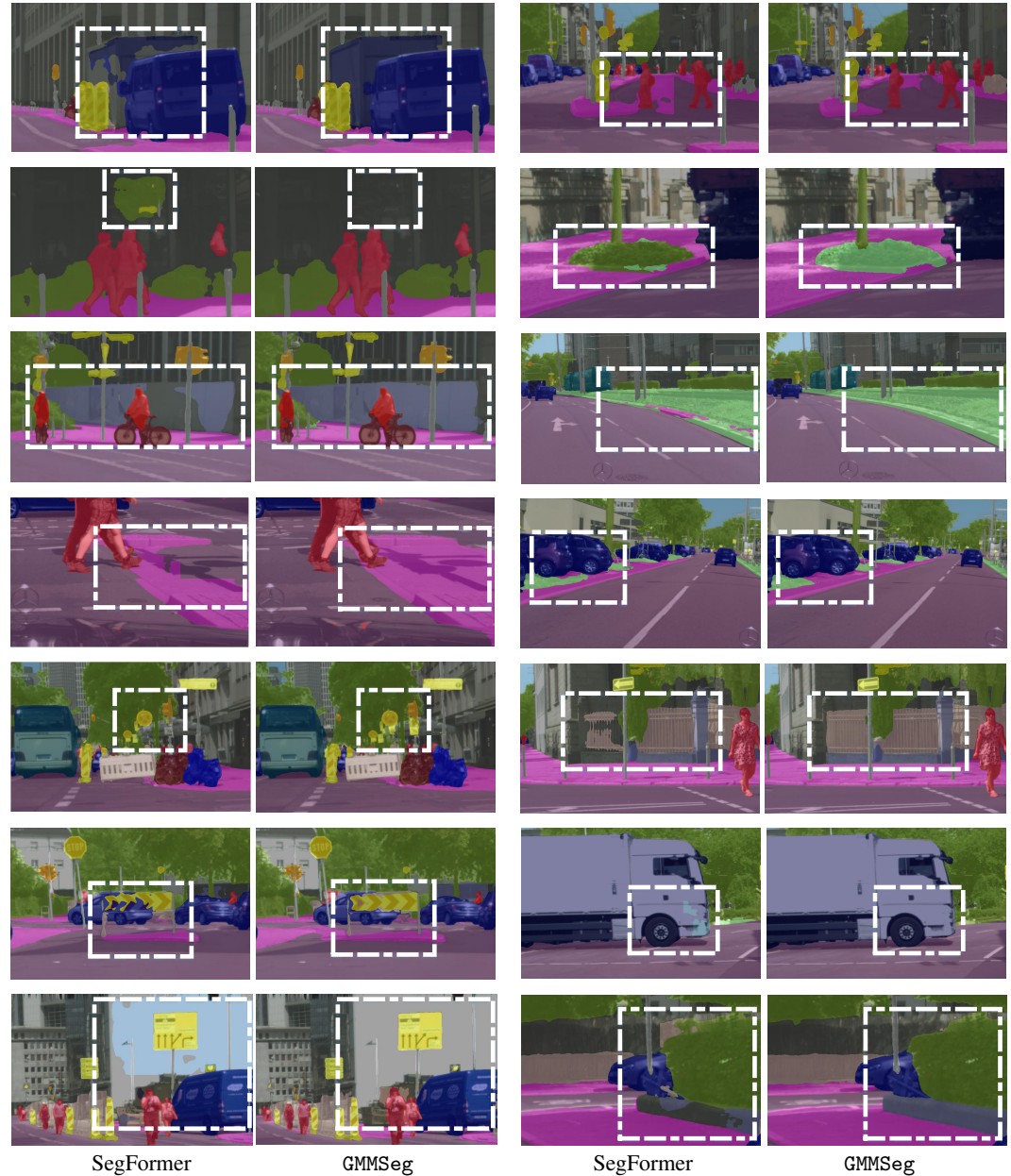

SegFormer      GMMSeg         SegFormer      GMMSeg

Figure S2: Qualitative results (§S3) of SegFormer [4] and our GMMSeg on Cityscapes [8].

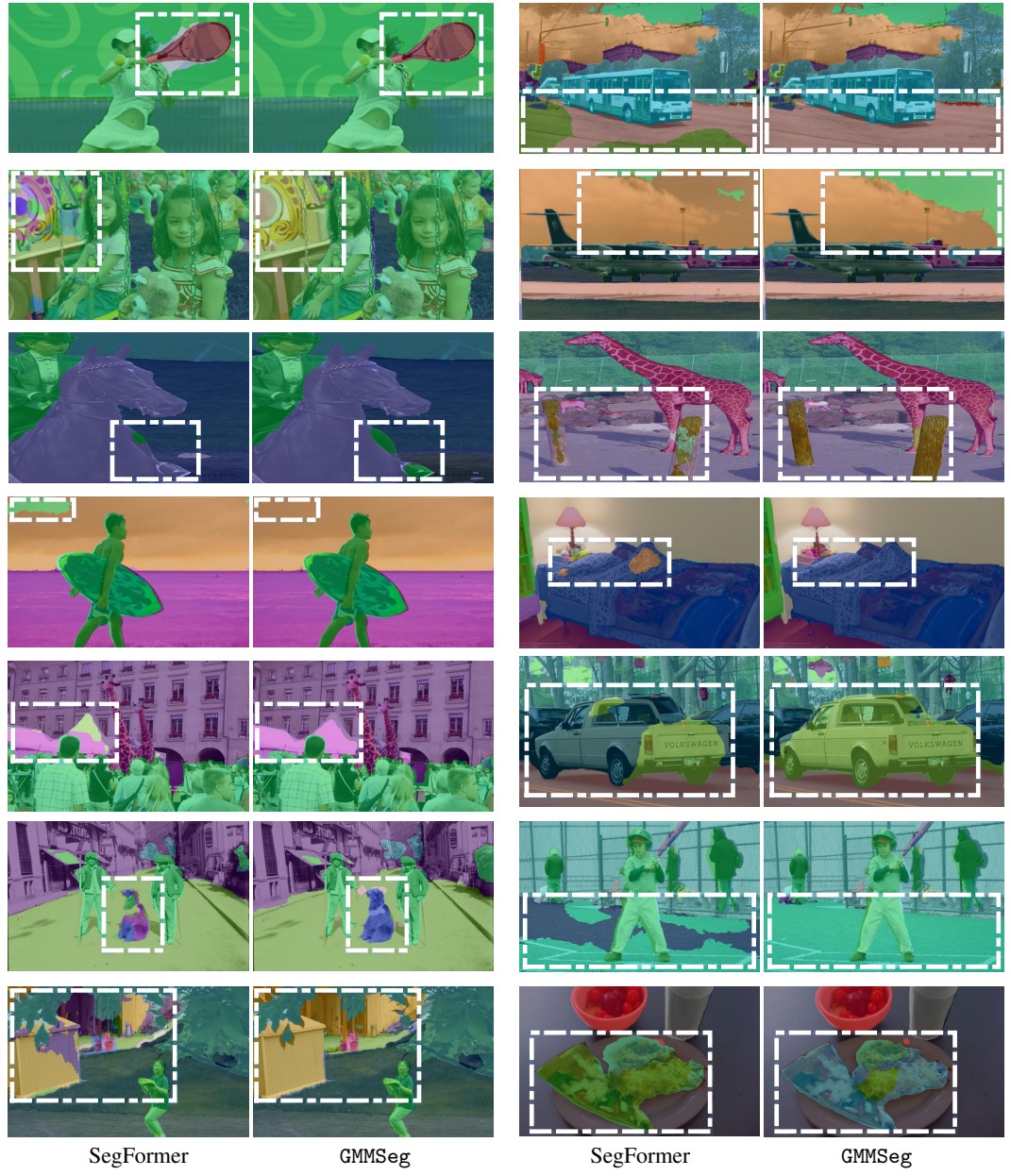

SegFormer  GMMSeg   SegFormer  GMMSeg

Figure S3: Qualitative results (§S3) of SegFormer [4] and our GMMSeg on COCO-Stuff [9].

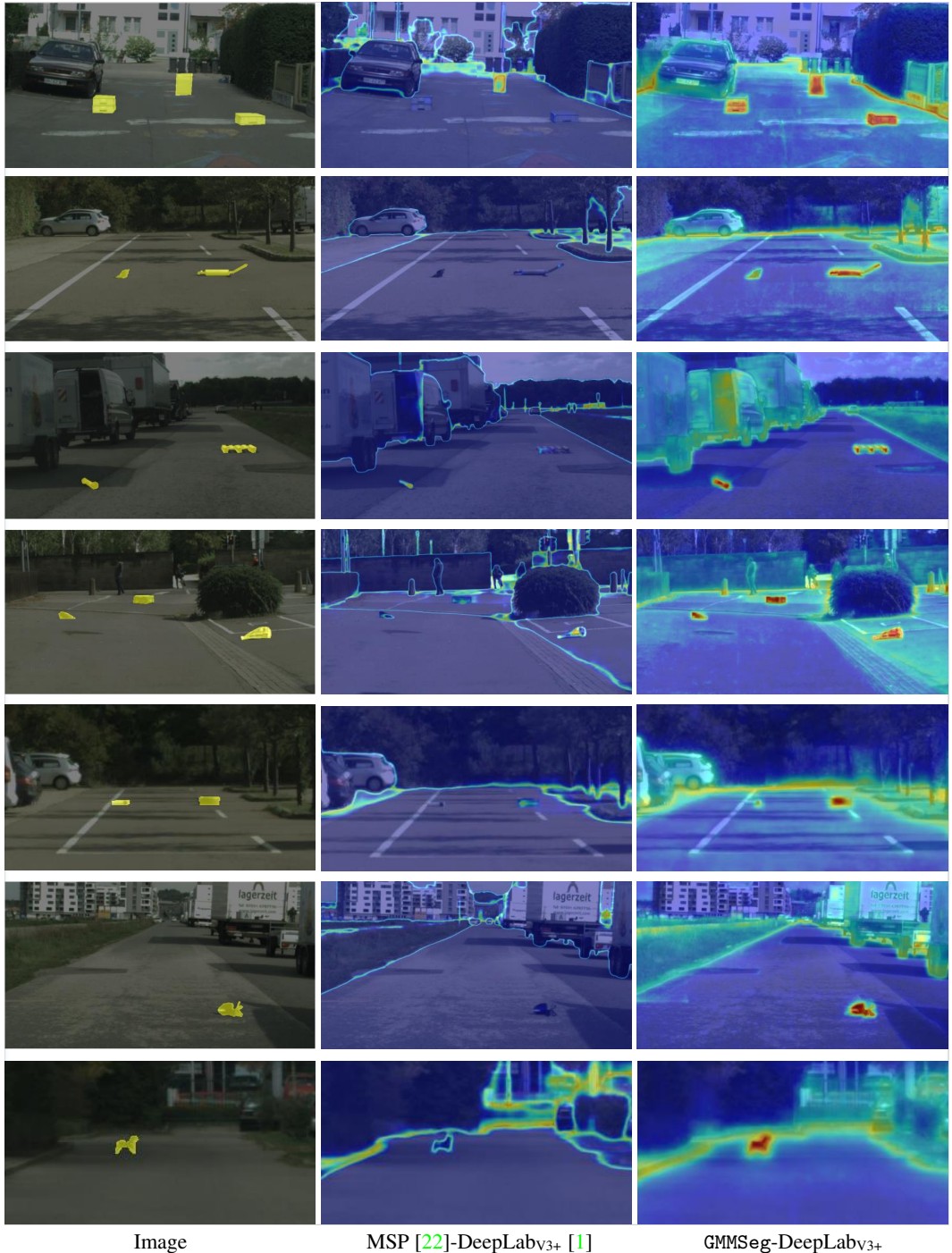

Image      MSP [22]-DeepLab$_{V3+}$ [1]     GMMSeg-DeepLab$_{V3+}$

Figure S4: Qualitative results (§S3) of anomaly heatmaps on Fishyscapes Lost&Found [12] val.