# OpenReview forum: "GMMSeg: Gaussian Mixture based Generative Semantic Segmentation Models"
_NeurIPS.cc/2022/Conference — NeurIPS 2022 Accept_

### Official Review · Reviewer_1Te2 · 2022-07-03

**Rating:** 7
**Confidence:** 4
**Soundness:** 4 excellent
**Presentation:** 3 good
**Contribution:** 3 good

**Summary:**

This paper proposes using a Gaussian Mixture Model (GMM) at the end of semantic segmentation models. The EM component maps the representations produced by a fully-convolutional neural network to the class labels and their probabilities. The GMM is learned simultaneously with the neural network; its parameters are optimized with Sinkhorn EM to fit the features from a running window over multiple training iterations. The authors show that this performs well both on classical semantic segmentation benchmarks and on "open world" tasks where anomalous/out-of-distribution objects are introduced in the test set.

**Questions:**

- After training, and in inference are the EM parameters simply those from the last training iteration? This'll still be based on a smaller memory bank, how far off from "ideal" EM parameters could these be? The supplement says that a large enough memory bank represents the overall distribution, is this assertions based only on the fact that the performance has saturated (up to 3 digits) at 32K?

### Minor comments/typos:

  * 68: Forth -> Fourth
  * 86: Suggest "modify [the] FCN architecture"
  * 88: "latter" or "later"?
  * 94: "modeling [the] underlying data distribution"
  * 181: Is "hard" the right adjective here? Maybe the grammar would be better with "...why it is hard for existing segmentation models to..."


**Limitations:**

The limitations seem to be discussed reasonably well. The submission focuses on a problem that is well-described by benchmarks, and the it demonstrates that the method is likely to perform better than competitors on out-of-distribution or anomalous examples.


**Strengths And Weaknesses:**

## Strengths

### i) Produces a generative, probabilistic model of the CNNs representations and the output classes.

A key part of the segmentation model, the final mapping from neural representations to class probabilities and labels, is done with an interpretable and well-calibrated model. A lot of value comes from the model producing a probability distribution, for interpretability, reliability, calibration, etc. Some reasonable estimation of uncertainty is important when using neural networks within a larger system (e.g. in a commercial product) where one might need to handle uncertain predictions differently, as opposed to the neural network being used in isolation (e.g. to maximize scores on a benchmark). This work contributes significantly to literature focused on achieving good uncertainty estimates in neural-net-based-models [a,b,c], and this is only one aspect of the contributions.

Using a GMM in this way produces a well-calibrated uncertainty estimate. The authors allude to this with the 3rd limitation of softmax described in the introduction. Reliability diagrams are given in the supplement (Fig S1).

### ii) Good choice of techniques for jointly training a generative model with a CNNs, well-validated by ablation studies.

Adding new types of layers, including ones inspired by non-deep-learning models, is a common subject of a paper. This work does a more principled treatment of this than usual. First, the GMM model is learned with a choice of algorithm (Sinkhorn EM) that will produce consistent results in few iterations, which seems especially important when this model fit is repeated throughout the overall SGD training. Use of a memory bank is also a good improvement to this.

The value of Sinkhorn EM is validated by ablation study. The impact of the memory bank is also studied in the supplement.

### iii) Good performance on anomaly detection.

The scores reported in the submission would not put it at the top of Fishyscapes leaderboard (https://fishyscapes.com/results). For example, as of writing the best AP score on the Fishyscapes lost & found is 63.43, vs 43.47 reported by the submission in Tables 2 & 3. However, this seems quote good for a method that isn't directly trained for anomaly detections. like the top published results in Fishyscapes. For instance, the current Fishyscapes leader [d] introduces synthetic anomalies when training a network. Instead, in the submission good anomaly detection can be extracted from a GMMSeg model trained on ordinary segmentation tasks.

This seem like a good demonstration of interpretability. As much as "interpretability" is an ambiguous term, the ability to infer new information from the model to use in other contexts seems like a useful example.

## Weaknesses

### iv) Less clear that it's an improvement over using CRF-based models in the same place, or why.

The authors contrast their against using softmax directly on the CNN features. They provide good reasons why GMMSeg would better model this problem. It seems like CRFs share a lot of the same advantages, and the contrast between GMMSeg and those methods is less obvious.

The original V1 of DeepLab [2], and several follow-up works [e,61] use CRFs to map the NN representation to class probabilities. These are cited in the related works, but the probabilistic modelling aspect of these works is not really discussed. Experimental comparisons against DeepLab are done with V3+ [47], a more recent version that dropped the CRF component.

GMMSeg also lacks the "context" priors that CRFs give, due to the pairwise (or higher-order) potentials included in those models.

### v) Not a fully combined optimization of EM and feature extractor.

In Section 3.2, the authors clarify:

> the extractor’s parameters θ are only updated by the gradient back-propagated from the discriminative loss, while the GMM’s parameters are only optimized by EM

I'd interpret this to mean that the feature extractor, which remains an important part of the model in GMMSeg, is still trained in a discriminative way that doesn't differ from previous methods. And the NN features are not optimized for input to the GMM component.

Given that this is the case, it's unclear why the EM needs to be learned at the same time as the feature extractor. Would there be any change to the training of the feature extraction NN if the GMM was simply ignored? And if so, could the GMM be learned *after* the NN is fully trained, without any change in results?

The authors make good design choices for an end-to-end joint optimization (iii), but stop short of doing so. However, empirical results do seem to show that the GMM still describes a better mapping from the NN features to the class labels than the original softmax that the NN was trained with.

### vi) Only considers segmentation.

All of the authors' justifications for using a GMM to map representations to class labels seems like they'd apply equally well to classification, or any deep network trained with softmax. I wonder why the paper focuses on image segmentation exclusively, and it does make it a less widely-impactful work.

### vii) Unclear presentation of feature extraction component.

Adjacent pixels are usually important in segmentation, the feature in the classification problem is not really the $x_n \in R^3$ as described in Section 3.1. The same is true for the formulation of the "dense feature extractor." I'd emphasize that the feature extractor really maps $R^{H \times W \times 3}$ to $R^{H \times W \times D}$.

[a] Oh et. al. "Modeling uncertainty with hedged instance embedding" ICLR 2019. \
[b] Caesar et. al. "Joint Calibration for Semantic Segmentation" BMVC 2015. \
[c] Nado et. al. "Uncertainty Baselines: Benchmarks for Uncertainty & Robustness in Deep Learning" \
[d] Grcić et. al. "Dense anomaly detection by robust learning on synthetic negative data" \
[e] Vemulapalli et. al. "Gaussian Conditional Random Field Network for Semantic Segmentation" CVPR 2016.

---

> ### Author Response · Authors · 2022-08-02
> **Response to Reviewer 1Te2 (Part I)**
>
> Thank you for recognizing the promising aspect of our work and providing valuable suggestions to help us improve clarity. We reply to the concerns and questions below. Our responses shall be incorporated into the revision.
>
> #### **Q1. Comparison and relation to CRF-based methods**
>
> **A1:** Thanks for your careful review.
>
> First, although CRF shares some advantages with GMMSeg, previous CRF-based segmentation models are still largely built upon discriminative softmax classifier, and cannot produce well-calibrated uncertainty estimations, hence struggling for handling OOD. For example, CRF-based models [2,61,e] contain the unary energy component, which is commonly modeled with negative label assignment probability $p(c|x)$ (model output after the softmax layer) [2,e] or logits (model output before the softmax layer) [61]. CRF-based models still rely on the softmax to parse the probability in a discriminative way. The 'sin' of softmax remains in these models (*c.f.*, Ln31-Ln36). From this aspect, GMMSeg earns superiority.
>
> Second, CRFs contain pairwise energy components, which model the pairwise priors as you mentioned. From this aspect, GMMSeg lacks the explicit probabilistic modeling of "context". However, modern network designs for segmentation models have already implicitly or explicitly captured the correlations among pixels during deep feature extraction (*i.e.*, CNN for gathering small local context, ASPP, and neural attention for long-range modeling). As a very early step towards a generative model based on GMM for image segmentation, our work also comes with a few intriguing questions, and this issue is one of them.
>
> Third, to better address your concern, we provide the comparison experiments on top of DeepLabV3+-ResNet101, on the ADE20K dataset as summarized in the below table. As seen, GMMSeg boosts performance. However, CRF post-processing even brings a negative impact. This is also one of the reasons that CRF post-processing is less used in current high-performance segmentation models.
>
> |                                     | mIoU |
> | ----------------------------------- | ---- |
> | DeepLab$_{\text{V3+}}$              | 44.6 |
> | DeepLab$_{\text{V3+}}$ + CRF        | 44.1 |
> | **GMMSeg-DeepLab$_{\text{V3+}}$**       | **46.0** |
>
> ---
>
> #### **Q2. Hybrid training of discriminative feature extractor and generative classifier**
>
> **A2:** Sorry for this misunderstanding.
>
> First, an appealing characteristic of our GMMSeg is that, the feature extractor is trained in a discriminative way, but the computation of the discriminative learning loss (Eq. 11) relies on the generative GMM classifier. Or more specifically, the posterior $p(c|x)$ is derived from the GMM. In turn, the EM optimization of GMM is conducted on the learned feature embedding space. Thus our method achieves a dense blend of generative optimization and discriminative training; the optimizations of EM and feature extractor are densely coupled. This allows our GMMSeg, as we repeatedly mentioned in the manuscript, to inherit the advantages of the two worlds.
>
> Second, as optimizations of EM and feature extractor are densely coupled, GMM classifier and feature extractor are both gradually updated and aligned with and adaptive to each other, making GMMSeg a compact model.
>
> These are also two crucial reasons why we cannot learn the GMM after the feature extractor is fully trained.
>
> Moreover, if the GMM was simply ignored, that means a softmax classifier is still needed to train the feature extractor. Actually, we have already conducted experiments with such a naïve strategy, at the very beginning of this project. We observed a clear performance drop, even compared with the original softmax-based baseline. We provide the experimental results below. We denote learning GMMs from a fully trained (with softmax) NN space as `DeepLabV3+ + GMM`.  Compared to `DeepLabV3+ + GMM`, an end-to-end trained GMMSeg-DeepLab$_{\text{V3+}}$ shows a significant improvement, which in turn supports our claim in Ln65-66.
>
> |                               | mIoU |
> | ----------------------------- | ---- |
> | DeepLab$_{\text{V3+}}$        | 44.6 |
> | DeepLab$_{\text{V3+}}$ + GMM  | 31.6 |
> | **GMMSeg-DeepLab$_{\text{V3+}}$** | **46.0** |
>
>
> The joint optimization of the generative classifier and discriminative feature embedding is one of the most exciting parts and contributions of this work. And, this is one of the keys to our promising performance. First training a feature extractor and then replacing the softmax classifier with a GMM cannot get good results.
>
> ---

---

> > ### Comment · Reviewer_1Te2 · 2022-08-09
> > **Seems good**
> >
> > This response clarifies some big things, and I continue to recommend acceptance.
> >
> > I'm still not sure I follow exactly how joint training is being done, but certainly glad that it is, and I think this comments gives some more hints. Based on this rebuttal, I think there's a good chance that it will be clearer in the final version.

---

> ### Author Response · Authors · 2022-08-02
> **Response to Reviewer 1Te2 (Part II)**
>
>
> #### **Q3. Extension to image classification**
>
> **A3:** Totally agree. The main reason is due to our limited GPU resources. But luckily, we just got some GPUs and produced some results. From our current very initial results, we indeed observe consistent performance improvement on top of the VGG, on the standard ImageNet dataset, by training from scratch. We thus believe our idea is powerful and principled. Extending GMMSeg to image classification is definitely our next focus.
>
>
> ---
>
> #### **Q4. Clarify presentation of feature extraction**
>
> **A4:** Thanks for your careful review! In the original submission, we keep notations simple, straightforward, and easy to follow. Some innocuous simplifications are also introduced to fit the page limit. After reading your comment, we feel the related statement may cause misleading. We will clarify the writing with a footnote:
>
> In segmentation, dense feature extractor $f_{{\theta}}$ maps pixel samples with image context, *i.e.*, $f_{{\theta}}: \mathcal{R}^{H\times W\times 3}\rightarrow \mathcal{R}^{H\times W\times D}$. We simplify the notations into pixel form, *i.e.*, $f_{{\theta}}: \mathcal{R}^{3}\rightarrow \mathcal{R}^{D}$, to keep a straightforward formulation.
>
> ---
>
>
> #### **Q5. Questions on 'Ideal' EM parameter estimation with memory**
>
> > Q5.1. After training, and in inference are the EM parameters simply those from the last training iteration?
>
> **A5.1:** Yes.
>
> > Q5.2. This'll still be based on a smaller memory bank, how far off from "ideal" EM parameters could these be?
>
> **A5.2:** The estimated parameters are not only based on the memory, but moving statistics over the entire training dataset thanks to the momentum EM mechanism (*c.f.*, Ln248-Ln251). The momentum mechanism stabilizes the training (*c.f.*, Ln251-252) and also helps us collect statistics over the whole dataset without a complicated post-process.
>
> > Q5.3. The supplement says that a large enough memory bank represents the overall distribution, is this assertion based only on the fact that the performance has saturated (up to 3 digits) at 32K?
>
> **A5.3:**  Thanks for your careful review. Our assertion is NOT only based on the fact that the performance has saturated at 32K. It is commonly agreed that there is much redundant information for this pixel classification task, *i.e.*, many neighbor pixels are very similar. This is also verified by some sampling based training strategies in this field, such as [ref1] PointRend: Image Segmentation as Rendering.
>
> But this assertion might need careful examination; if we can have a large enough memory that can store the whole training dataset, maybe we can achieve better performance. However, this further brings the concern on the trade-off between performance and training resource cost.
>
> ---
>
>
> #### **Q6. Minor comments/typos**
>
> **A6:**  We are grateful for your careful review! We apologize for these errors and will definitely correct them.
>
> ---
>
> #### **Performance comparison with [d] "Dense anomaly detection by robust learning on synthetic negative data"**
>
> Thanks for summarizing the differences between our algorithm and [d], and admitting our contribution to the field of anomaly detection.
>
> Here we respectfully remind the reviewer that the comparison to [d], *i.e.*, 63.43, vs 43.47, is a little bit unfair. In addition to introducing synthetic anomalies during training, [d] adopts a trick (Sec. 5.2, page 14 of [d]): "... a variant of our method which focuses on the ground during inference (GF stands for ground focus). We define the ground as a convex hull which covers all pixels predicted as the class road or the class sidewalk. This change entails a huge improvement in both metrics". We personally do not admire such a strategy, as this is just using the data bias of the Fishyscapes dataset. It is not held in general cases: the anomaly may appear everywhere in the view.
>
> Without such GF trick,  [d] only gains 39.4 AP (Table 2), which is inferior to ours (43.47). Without neither architectural change (like [34–37]), nor re-training (like [38–40]), nor post-calibration (like [17,18,41–46]), our algorithm yields very promising results on anomaly detection. We believe our work brings fundamental insights into this field. And, as you noticed, this is only one aspect of our contributions.
>
> ---
> Finally, thank you again for your very detailed and constructive comments, from which we really learn something!

---

### Official Review · Reviewer_TvcJ · 2022-07-12

**Rating:** 5
**Confidence:** 4
**Soundness:** 3 good
**Presentation:** 3 good
**Contribution:** 2 fair

**Summary:**

This paper proposes GMMSeg, a generative model based on GMM for image segmentation that models the joint distribution of pixel features and classes. For each class, GMMSeg builds GMMs via EM so as to capture class-conditional densities. A deep network is trained end-to-end in a discriminative manner to extract features for the GMMs. This endows GMMSeg with the strengths of both generative and discriminative models. With a variety of segmentation architectures and backbones, GMMSeg outperforms the discriminative counterparts on three closed-set datasets, ADE20K, Cityscapes and COCO. Without any modification, GMMSeg even performs well on open-world datasets.

**Questions:**

Please refer to the Strengths And Weaknesses section

**Limitations:**

The authors addressed the limitations and potential negative societal impact of their work

**Strengths And Weaknesses:**

Strength:
1. It is reasonable to use generative classification models with discriminative feature learning to combine the strengths of both generative and discriminative models.
2. The properties of GMMs make GMMSeg well adapt to multimodal data densities and allows GMMSeg to naturally reject abnormal inputs.
3. GMMSeg is fully compatible with different network architectures.
4. Extensive experiments are conducted on multiple benchmark datasets. It shows that the proposed method improves the performances of multiple baselines.

Weakness:
1. The proposed generative framework is complex. The motivation of using GMMs on top of a deep network is not very clear. Why not directly model the joint distribution of image and label maps using deep networks?
2. The meaning of the components of GMMs in the proposed framework is not very clear. It looks like all classes have the same number of components and every component is expected to have the same amount of pixels. However, the modality of different classes and the pixels of different components are not always the same.
3. It would be more convincing if there are some discussion on what each components represent and a visualization of the learned components.
4. Compared with the baseline, GMMSeg brings more overheads at both inference and training time. There should be a comparison in terms of training/inference speed. it will be more convincing if the proposed method is compared with baseline+CRF post processing, as both GMMSeg and CRF improve performance at the cost of computation overhead.
5. It is oversimplified to model segmentation as pixel classification using GMMs as it does not handle the correlations among pixels in an image. It might be more reasonable to use MRF, which has been explored in previous work, e.g. [a]
[a] Liu, Ziwei, et al. "Deep learning markov random field for semantic segmentation." IEEE transactions on pattern analysis and machine intelligence 40.8 (2017): 1814-1828.

---

> ### Author Response · Authors · 2022-08-02
> **Response to Reviewer TvcJ (Part I)**
>
> Thank you for your time and valuable feedback. Below we address your concerns with additional experiments and discussions. We shall incorporate the responses into the revision.
>
> #### **Q1. ''Why not directly model the joint distribution of image and label maps using deep networks?''**
>
> > Q1.1. The motivation of using GMMs on top of a deep network is not very clear.
>
> **A1.1:** GMMs can express almost arbitrary continuous distributions and is among the most famous probabilistic generative models (Ln114). The hybrid framework, *i.e.*, using GMMs on top of a deep representation embedding network, endows GMMSeg with the strengths of both discriminative and generative models:
>
> 1. The discriminative representation extractor offers *expressive* feature representations as demonstrated by extensive experiments in Sec. 4.1.
> 2. The generative classifier allows GMMSeg to capture the multimodality of data, to be well-calibrated, and to naturally reject the abnormal inputs without any modification, as demonstrated by experiments in Sec. 4.2.
>
> Please also refer to Ln58-Ln60, Ln63-69 in the Introduction section, and Ln103-121 in the Related Work section of the main paper, and Sec. S2 in the *suppl.* for detailed discussions on our motivation.
>
> Ln58-Ln60: *'GMMSeg smartly learns generative classification with end-to-end discriminative representation in a compact and collaborative manner, exploiting the benefit of both generative and discriminative approaches.'*
>
> Ln63-69: *'... with the hybrid training strategy – online EM based classifier optimization and end-to-end discriminative representation learning, GMMSeg can precisely approximate the data distribution over a robust feature space... the distribution preserving property allows GMMSeg to naturally reject abnormal inputs, without neither architectural change (like [34–37]) nor re-training (like [38–40]) nor post-calibration (like [17, 18, 41–46]).'*
>
> Ln103-121: *'... discriminative classifier is used exclusively [24], due to its simplicity and excellent discriminative performance ... generative classifiers are widely agreed to have several advantages ... accurately modeling the input distribution, and explicitly identifying unlikely inputs in a natural way'*
>
> > Q1.2. Why not directly model the joint distribution of image and label maps using deep networks?
>
> **A1.2:**
>
> 1. Directly modeling the joint distribution of very high-dimensional image data and label maps is more challenging, which is even a little beyond the scope of this work. We would like to take this as a part of our future work.
>
> 2. GMMSeg is fully compatible with modern segmentation (dense feature extraction) network architectures. It can replace the discriminative softmax seamlessly, and thus can take the full benefits from the development of segmentation network architecture design (*i.e.*, dense feature extractor).
>
>
> ---
>
> #### **Q2. Clarification of GMM component**
>
> > Q2.1. The meaning of the components of GMMs in the proposed framework is not very clear.
>
> **A2.1:** Each mixture component can be viewed as a representative pattern that resides in the feature space, discovered in an EM, data-driven manner. Please also refer to `Q3. Visualization of learned components` for illustrative examples.
>
> > Q2.2. It looks like all classes have the same number of components.
>
> **A2.2:** We want to avoid introducing delicate designs like this to make our framework simple, general, and elegant. And we also show such a simple strategy is enough to achieve impressive performance. Actually, it is quite simple for us to use different numbers of components for different classes, based on the occurrence frequency of class samples.
>
> In addition, the widely used discriminative softmax classifier only learns one single weight vector for each class, totally ignoring the multimodality nature of data and the task. Considering our method makes an initial step towards a generative mixture model based on GMM for image segmentation, and its good properties of multimodality modeling, treating “all classes have the same number of components” as a weakness of our work seems unfair.
>
> > Q2.3. ... and every component is expected to have the same amount of pixels.
>
> **A2.3:** Sorry for this misunderstanding. Here we clarify that the equipartition assumption is *NOT* a hard constraint. It is totally OK (and always) that different clusters are assigned with different numbers of data samples. The equipartition assumption, widely used in clustering, is just a soft constraint for avoiding the degenerate solution, *i.e.*, all data samples are partitioned to a single cluster, as pointed out by [107,108].
>
> ---

---

> ### Author Response · Authors · 2022-08-02
> **Response to Reviewer TvcJ (Part II)**
>
>
> #### **Q3. Visualization of learned components**
>
> **A3:** We visualize the learned components in Sec. S2 in the revised *suppl.*. It shows the probability of pixel assignments with $M=3$ mixture components for each class. Different components are illustrated by different colors (*i.e.*, red, green, blue). For each pixel, the highest probability of being assigned to the component is visualized using the corresponding color. As shown in Fig. S1, GMMSeg can automatically discover informative patterns in the class.
>
> ---
>
> #### **Q4. Overhead in training/inference**
>
> **A4:** Sorry for this confusion. We have discussed these issues. In the Limitation Analysis section (*c.f.*, Sec. S4 in the *suppl.*), we report the training computational overhead:
>
> > One limitation of our approach is that the EM based generative parameter estimation needs extra optimization loops in each training iteration which would reduce the training efficiency in terms of time complexity. However, in practice, we find one EM loop per training iteration is good enough for global model convergence, which only brings a minor computational overhead, *i.e.*, ~5% training speed delay.
>
> For the inference, the delay is almost negligible (Ln278). We report the detailed inference speed in the below table. The same setup to the ablations is adopted, where the DeepLab$_{\text{V3+}}$-ResNet101 architecture is used. Speed is measured on a single NVIDIA GeForce RTX 3090 GPU.
>
> |                               | fps   |
> | ----------------------------- | ----- |
> | DeepLab$_{\text{V3+}}$        | 14.16 |
> | GMMSeg-DeepLab$_{\text{V3+}}$ | 13.37 |
>
> ---
>
> #### **Q5. Comparison to CRF post-processing; ''*Both GMMSeg and CRF improve performance at the cost of computation overhead*''**
>
> **A5:** First, GMMSeg introduces minor computational overhead at training (~5% training speed delay). Once trained, the model shows comparative inference speed (*c.f.*, `Q4. Overhead in training/inference`) to the counterpart.
>
> Second, to the best of our knowledge, GMMSeg is the first semantic segmentation method that reports promising results on both closed-set and open-world scenarios by using a single model instance. It shows exciting results on anomaly segmentation where the discriminative CRF approaches suffer. We believe GMMSeg brings orthogonal contributions to the community.
>
> Third, to address your concern, we provide below the comparison results on top of DeepLab$_{\text{V3+}}$-ResNet101, on the ADE20K dataset. As seen, GMMSeg yields much better performance, compared with CRF post-processing. The latter even brings a negative impact. This is also one of the reasons that CRF post-processing is less used in current high-performance segmentation models.
>
> |                               | mIoU |
> | ----------------------------- | ---- |
> | DeepLab$_{\text{V3+}}$        | 44.6 |
> | DeepLab$_{\text{V3+}}$ + CRF  | 44.1 |
> | **GMMSeg-DeepLab$_{\text{V3+}}$** | **46.0** |
>
> ---
>
> #### **Q6. Oversimplified to model segmentation as pixel classification; Missing modeling of pixel correlations**
>
> **A6:** First, many previous methods formulate semantic segmentation as *pixel classification* [1].
>
> Second, modern network designs for segmentation models have already implicitly or explicitly captured the correlations among pixels during deep feature extraction (*i.e.*, CNN for gathering small local context, ASPP, and neural attention for long-range modeling).
>
> Third, even with such an elegant model, consistent performance gains can still be observed, which exactly demonstrates the power of our idea and the novelty of this work.
>
> Fourth, as a very early step toward a generative model based on GMM for image segmentation, our work also comes with a few intriguing questions, and this issue is one of them.
>
> ---

---

> > ### Comment · Reviewer_TvcJ · 2022-08-09
> > **I read the authors' response and other reviewers' comment**
> >
> > I read the authors' response and other reviewers' comment and changed my rating to borderline accept

---

> ### Author Response · Authors · 2022-08-08
> **Further discussion**
>
> Dear reviewer,
>
> In our previous responses, we have tried our best to address your concerns.
>
> If you have any further comments or questions, please let us know. Thanks.

---

### Official Review · Reviewer_5AWr · 2022-07-15

**Rating:** 6
**Confidence:** 4
**Soundness:** 3 good
**Presentation:** 3 good
**Contribution:** 3 good

**Summary:**

This paper presents a novel paradigm for semantic segmentation. Specifically, the authors propose to frame the segmentation task as a generative model by modeling the probability of p(x|c) as a mixture of Gaussians. This is rather different from the popular discriminative model that adopts Softmax.

**Questions:**

- what about using a different p(c) ?

**Limitations:**

- the assumption of uniform distribution of p(c) is not very reasonable.

**Strengths And Weaknesses:**

+ interesting idea
+ novel paradigm, as compared to a large number of semantic segmentation models.

- the assumption of p(c) = 1/C might not be very reasonable. Normally, the prior distribution of different classes is not a uniform distribution.

---

> ### Author Response · Authors · 2022-08-02
> **Response to Reviewer 5AWr**
>
> Thank you for your encouraging comments. Below we address the raised point with additional discussions that we will include in the final version.
>
> #### **Q. The assumption of uniform distribution of p(c)**
>
> **A:** Here the uniform prior is adopted because people usually do not have a strong belief on the class distribution beforehand, as we cannot identify the prior distribution of infinite incoming data in real-world applications [ref1]. On the other hand, the uniform prior is quite simple and widely used in related fields [ref2].
>
> But we also agree that some other choices like counting the class frequency in the training set as a prior should be investigated. However, the study of alternative class priors is a little beyond the scope of this paper and would potentially have a negative impact on comparison fairness (as most previous segmentation methods implicitly adopt the uniform prior).
>
> To better address your concern, we report below the performance obtained by using the class frequency in the training set as the prior. We observe a slight performance drop, *i.e.*, - 0.4% in terms of mIoU with DeepLab$_{\text{V3+}}$-ResNet101 architecture on the ADE20K dataset.
>
> |                                                          | mIoU |
> | -------------------------------------------------------- | ---- |
> | GMMSeg-DeepLab$_{\text{V3+}}$ (Uniform Prior)             | 46.0 |
> | GMMSeg-DeepLab$_{\text{V3+}}$ (Training Occurrence Prior) | 45.6 |
>
>
> As a very early step towards a generative model based on GMM for image segmentation, our work comes with a few intriguing questions, and this issue is one of them.
>
> [ref1] Bayesian data analysis. In Chapman and Hall/CRC, 1995.
>
> [ref2] A Bayesian Hierarchical Model for Learning Natural Scene Categories. In CVPR05.
>
> ---

---

### Official Review · Reviewer_o8w9 · 2022-07-16

**Rating:** 5
**Confidence:** 4
**Soundness:** 3 good
**Presentation:** 2 fair
**Contribution:** 2 fair

**Summary:**

This paper studies semantic segmentation. The authors developed a Gaussian Mixture-based Generative Semantic Segmentation Model. The experimental results on serval datasets demonstrate the effectiveness of the proposed method.

**Questions:**

What's the effect of the proposed method on the model inference speed?

**Limitations:**

Yes

**Strengths And Weaknesses:**

[Strengths]
+ Compared to the baselines, the proposed methods could bring constant improvements
+ The abundant experiments are introduced to prove the effectiveness of the proposed method.

[Weaknesses]
- some important references are missing.
[a] Top-down Learning for Structured Labeling with Convolutional Pseudoprior, ECCV 2016.
[b] Exploring Cross-Image Pixel Contrast for Semantic Segmentation, ICCV 2021.

- Both [a] and [b] introduce data distribution p(pixel feature|class), they all need to be included in the discussion. [b] also learn the class-related feature and bring constant improvements. It could be better to add it into comparison.

---

> ### Author Response · Authors · 2022-08-02
> **Response to Reviewer o8w9**
>
> Thank you for your time and valuable feedback. All the comments and questions are addressed below. We will incorporate our responses in the revision.
>
> #### **Q1. Missing reference [a, b]**
>
> **A1:** Thanks for bringing these two excellent works to our attention. We are happy to discuss and compare [a, b] in our final version. Though both [a] and [b] consider data distribution p(*pixel feature*|*class*) to some extent, they still rely on the *discriminative* softmax classifiers. In addition, [a] requires three fragile phases for training. However, our algorithm discards softmax classifiers from the beginning. More importantly, our method demonstrates it is indeed possible to train simultaneously a generative classifier with deep feature representation, and even shows better performance in both closed-set and open-world settings.
>
> ---
>
> #### **Q2. Effect of inference speed**
>
> **A2:** The impact of inference speed is almost negligible (Ln278). We keep the same setup to the ablations, where the DeepLab$_{\text{V3+}}$-ResNet101 architecture is used. Speed is measured on a single NVIDIA GeForce RTX 3090 GPU.
>
> |                               | fps   |
> | ----------------------------- | ----- |
> | DeepLab$_{\text{V3+}}$        | 14.16 |
> | GMMSeg-DeepLab$_{\text{V3+}}$ | 13.37 |
>
>
> ---

---

### Meta-Review · Area_Chair_USjb · 2022-08-26

**Recommendation:** Accept
**Confidence:** Less certain

**Metareview:**

This paper proposes to learn generative model (mixture of Gaussian) on the discriminative features. The proposed method achieves strong performance on semantic segmentation and it is capable of anomaly detection.

The paper was reviewed by 4 reviewers.

Reviewer o8w9 (rating: 5) pointed out 2 missing references and asked about speed. The authors clarified the difference between their work and the two references, and showed that the inference speed is negligible.

Reviewer 5AWr (rating: 6) asked about assumption of uniform prior on classes. The authors explained that this is a common assumption, and added results on learned non-uniform class prior.

Reviewer TvcJ (rating: 5) asked about the motivation for the Gaussian mixture model, the meaning of the mixture components, computational overhead, comparison with MRF prior. The authors addressed the questions in detail and added new results. The reviewer read the authors' rebuttal and raised the rating to the current level 5.

Reviewer 1Te2 (rating: 7) wrote a very detailed review and was mostly satisfied with the authors' rebuttal, although this reviewer was still not entirely sure about joint training.

---------------

Overall, for the two reviewers with ratings 5, their concerns did not suggest serious flaws of the paper, and the authors addressed their concerns satisfactorily.

I thus lean toward accepting this paper.


**Award:**

No

---

### Decision · Program_Chairs · 2022-09-14

Accept